# All-in-one, all-optical logic gates using liquid metal plasmon nonlinearity

Jinlong Xu [1,2,6], Chi Zhang[2,6], Yulin Wang[2,3,6], Mudong Wang[2], Yanming Xu[1,2], Tianqi Wei[2], Zhenda Xie [2] ✉, Shiqiang Liu[2], Chao-Kuei Lee [4], Xiaopeng Hu [2] ✉, Gang Zhao[2], Xinjie Lv[2], Han Zhang [5], Shining Zhu [2] & Lin Zhou [2] ✉

Electronic processors are reaching the physical speed ceiling that heralds the era of optical processors. Multifunctional all-optical logic gates (AOLGs) of massively parallel processing are of great importance for large-scale integrated optical processors with speed far in excess of electronics, while are rather challenging due to limited operation bandwidth and multifunctional integration complexity. Here we for the first time experimentally demonstrate a reconfigurable all-in-one broadband AOLG that achieves nine fundamental Boolean logics in a single configuration, enabled by ultrabroadband (400–4000 nm) plasmon-enhanced thermo-optical nonlinearity (TONL) of liquid-metal Galinstan nanodroplet assemblies (GNAs). Due to the unique heterogeneity (broad-range geometry sizes, morphology, assembly profiles), the prepared GNAs exhibit broadband plasmonic opto-thermal effects (hybridization, local heating, energy transfer, etc.), resulting in a huge nonlinear refractive index under the order of $10^{-4}$–$10^{-5}$ within visual-infrared range. Furthermore, a generalized control-signal light route is proposed for the dynamic TONL modulation of reversible spatial-phase shift, based on which nine logic functions are reconfigurable in one single AOLG configuration. Our work will provide a powerful strategy on large-bandwidth all-optical circuits for high-density data processing in the future.

Computer technology based on high-speed logics is the cornerstone of modern information processing and communications. Nevertheless, current computer processors built by electronic circuits may confront the physical limitation to continue Moore's law within next two decades[1]. Future development of computer technology requires new principles and technologies. Photonic circuit is now widely regarded as one of the most potential successors to its electronic counterparts because optical signal processing has a series of advantages such as ultrahigh bit-rate, large bandwidth, great concurrency, as well as ultralow cross-talk[2–4]. Nowadays, the practical implementation of optical processing is based on optic-electric interconnection in which the digital signal is processed in electronic processor and the light is applied as signal transmitter. However, its performance is far from the speed ceiling of optical computation since the restriction of high-latency and cumbersome optic-electric conversion. In this respect, it is of great importance to progress all-optical system by replacing the

[1]Department of Physics, College of Physics and Information Engineering, Fuzhou University, Fuzhou, China. [2]National Laboratory of Solid State Microstructures, School of Electronic Science and Engineering, College of Engineering and Applied Sciences, Nanjing University, Nanjing, China. [3]Department of Physics, Nanjing Tech University, Nanjing, China. [4]Department of Photonics, National Sun Yat-sen University, Kaohsiung, Taiwan. [5]Key Laboratory of Optoelectronic Devices and Systems of Ministry of Education and Guangdong, College of Physics and Optoelectronic Engineering, Shenzhen University, Shenzhen, China. [6]These authors contributed equally: Jinlong Xu, Chi Zhang, Yulin Wang. ✉e-mail: xiezhenda@nju.edu.cn; xphu@nju.edu.cn; linzhou@nju.edu.cn

entire electronic functional components with all-optical elements[5,6]. The promising building blocks for optical processors are the AOLGs, which can enable logic functions by manipulating the intensity, phase, polarization, or wavelength of optical signals through light-matter interaction.

Among the intensive research of AOLGs in the past decades, bandwidth scalability and multifunctionality are among the most severe challenges determining the feasibility of high-speed optical processors[3,6]. The practicality of optical processors implies the urgent demand of multifunctional AOLG components since different optical logic gates will need to be made physically different. This will give different spatial modes or entail many components having different spatial mode requirements, now making each component bulky, unique, and not easily printable in a universal way like electronics. The past decade has witnessed vast advancements in AOLGs based on linear or nonlinear optical modulation, ranging from stimulated scattering and photoluminescence (PL) of nanowires or nanospheres[7–9], linear interference with phase filtering[1,10], to spatial self-phase modulation (SSPM) or spatial cross-phase modulation (SXPM) of nanosheets[11–13]. However, most of these strategies reported thus far have yet to show wide operation bandwidth (only lay within tens of nanometers) or superior compatibility of multifunctional integration, shadowing their potential for massively parallel processing. In this case, more rational strategies have been long pursued for the further development toward high scalability and multifunctionality.

The intriguing optical nonlinearity arisen from the surface plasmonics of nanostructured metallic materials provides a new approach to light manipulation in nano-optical devices[14–18]. As a special class of metal, room temperature liquid metal alloys typified by Galinstan and eutectic GaIn have been of great interest nowadays due to their stable liquid phase, exceptional stretchability, strong plasmonic effect, high thermal conductivity, and electrical conductivity, together with biocompatible low toxicity compared with mercury[19–21]. These unique properties imply valuable potential of liquid-metal-based plasmonic nanostructures for exploring advanced AOLGs, yet the corresponding strategy remains undiscovered.

In this work, we exploit the plasmon-enhanced TONL of liquid-metal GNAs to demonstrate a reconfigurable all-in-one AOLG based on dynamic and programmable manipulation on the reversible phase shift of dual-beam SXPM interaction. In such a single all-optical configuration without external electronic modulation, nine fundamental Boolean logics are achievable including AND, OR, NOT, NOR, NAND, XNOR, XOR, material implication (IMP), and not material implication (NIMP). The ultrabroadband (400–4000 nm) plasmon-enhanced light harvesting of GNAs endows a logic operation band ranging from visible to infrared regions. Our results would provide a promising strategy to overcome the limitation of bandwidth and multifunctionality in traditional AOLG schemes, inspiring a new pathway toward optical processor.

## Results

### Characterization of Galinstan nanodroplets
The broadband all-in-one AOLG platform is highly dependent on the intrinsic optical properties (dielectric constant, optical absorption, etc.)[22,23] of Galinstan liquid metals and their plasmonic thermo-optical responses (Fig. 1). The representative optical photograph and measured dielectric functions ($\varepsilon_r$: real part, $\varepsilon_i$: imaginary part) of the as-synthesized bulk Galinstan droplets are depicted in Fig. 1a–c, respectively. Note that Galinstan exhibits comparable $\varepsilon_r$ to the noble metal Au and Ag, while $\varepsilon_i$ of Galinstan is one or two orders larger, implying that Galinstan is a superior plasmonic material for broadband light harvesting with a much higher figure of merit ($-\varepsilon_i/\varepsilon_r$) with respect to conventional plasmonic metals[15] (see Supplementary Fig. 1 for details). These intriguing characteristics indicate versatile potential of liquid metal family in opto-thermal based optical applications.

Apart from the material priority of giant intrinsic absorption, pronounced heterogeneity of the plasmonic nanostructures of liquid metals is crucial as well. In order to obtain the liquid nanoparticles with broad distributed sizes, in the experiment, the GNAs were prepared by moderate ultrasonication exfoliation of bulk droplets in N-methyl-2-pyrrolidinone (NMP) solvent (see "Methods", Supplementary Section I and Supplementary Fig. 2 for details). The transmission electron microscope (TEM) image of the as-prepared GNAs (Fig. 1d) depicts a variety of particle sizes ranging from ~ 30 to 150 nm (Fig. 1e) and versatile assembly configuration, beneficial for high density of plasmonic modes and resonant interparticle coupling. The measured absorption spectrum of the GNAs in Fig. 1f clearly reveals an efficient localized-surface-plasmon (LSP) resonance capability with ultrabroadband light absorption across the visible to mid-infrared wavelength range (400–4000 nm), superior to most reported LSP based plasmonic absorbers[24–26].

The LSP absorption performance is well reproduced by the simulated results (dashed line in Fig. 1g, see more details in Supplementary Section II). Such exceptional performance can be attributed to several unique advantages from both material and nanostructure. (1) The ultrasonic initiated nanoparticles are highly heterogenous with wide-distribution sizes owing to the easy disintegration nature of liquid phase (Supplementary Fig. 3a, b), forming extremely high density of optical modes (ideal for a broadband spectral range of LSPs). (2) The excellent inherent stretchability of liquid phase makes the GNAs tend to be morphed into massive irregular geometries under stress and gravity[27] (Supplementary Fig. 3c). It leads to strong resonant interparticle hybridization with local field enhancements and additional new resonant mode formation[28], which further enhance and broaden the LSP response. (3) Profiting from the self-limiting oxidation of gallium, the prepared GNAs are conformably wrapped with insulation nanolayers (~2 nm thickness) of $Ga_2O_3$[29,30], forming a self-assembling Galinstan/$Ga_2O_3$ core-shell framework (see inset of Fig. 1e). The self-limiting oxide layers can effectively prevent electron transfer among adjacent nanodroplets, providing a robust shield for keeping strong surface electron localization and stable plasmonic effects. As a result, such pronounced plasmonic optical absorption is crucial to enable strong interfacial opto-thermal effect around the liquid metal nanoparticles, beneficial for arising ultrabroadband-responsive plasmonic nonlinear optics such as the plasmon-induced resonant energy transfer (PIRET)[31–33] and plasmon-enhanced TONL modulation[34].

The broadband plasmon-enhanced TONL of the prepared GNAs is then measured by quantifying the SSPM effect based on the PIRET process, of which the experimental setup, mechanism, and results are demonstrated in Supplementary Fig. 7a, b and Fig. 1g, h. In the experiment, when the GNAs are excited by a laser beam, massive hot electrons are generated by intraband transition of conduction electrons at ground states below the Fermi level ($E_F$), followed by non-radiation electron-phonon scattering with the PIRET process across GNA/NMP interfaces (bottom panel of Supplementary Fig. 7a). The localized plasmonic energy leads to highly localized thermal distribution inside nanodroplets, followed by a strong thermal-induced nonlinear refractive index field (RIF) generated in the locally surrounding NMP with Gaussian-like diffusion. It finally leads to a spatial mode transition of propagation beam from input fundamental Gaussian beam to a pattern of $N$-order diffraction ring mode $N = (n + 1)/2$ ($n = 1,3,5...$), when the phase-shift between the center and the edge of beam cross section reaches $n\pi$ (Detailed descriptions are given Supplementary Section III). Consistent with the broadband plasmonic absorption, strong diffraction ring modes are experimentally observed under laser excitation of different wavelengths (top panel of Fig. 1g). These patterns match well with the theoretical simulations based on Kirchhoff's diffraction integral[35] (bottom panel of Fig. 1g, see Supplementary Section IV for details), confirming the spatial-phase shift by

SSPM. The effective-nonlinear refractive index $n_2$ of the prepared GNA samples can thus be extracted from these results (as shown in Fig. 1h), which reaches up to orders of magnitude -$10^{-4}$–$10^{-5}$ for a wide range of excited wavelengths (400–3500 nm) (Detailed results are given in Supplementary Table S1). Most strikingly, one can find that, the $n_2$ value of GNAs is among the highest level across extremely broad wavelength range as compared to the state-of-art results of typical nanomaterials[36–46] (Fig. 1h and Supplementary Table S1), exhibiting the vast potential for efficient nonlinear light manipulation at large bandwidth and parallelism.

## Structure and mechanism of all-optical logic gate

By taking full advantage of the broadband-responsive TONL, we further demonstrate that the GNA samples can be ideal candidates for large-bandwidth and multifunctional AOLGs through programmable control of the reversible spatial mode transition. A proof-of-concept

AOLG configuration was designed and established for arbitrary 2-bit logic processing based on a dual-beam SXPM modulation at the plasmonic TONL, as schematically shown in Fig. 2a. Three GNA dispersion samples sealed in 1-mm-thin cuvette were cascaded, with two samples (GNA-A and GNA-B) functioning as the 2-bit generators through the SSPM tuned by two independent laser beams (laser-A and laser-B). The two beams were then non-collinear focused into GNA-C with an angle of ~3-10° to implement SXPM operation. We used laser-A and laser-B as the control and signal beam in SXPM operation, respectively, with the initial power of laser-A higher than laser-B. Basically, the control beam generates and modulates the RIF within GNA-C, bringing multiple variation of signal-beam phase shift to implement multifunctional AOLG processing. More detailed mechanism will be explicated in the later section.

As illustrated in Fig. 2b, we define the fundamental Gaussian mode with bright far-field center as logical state '1' and first-order diffraction

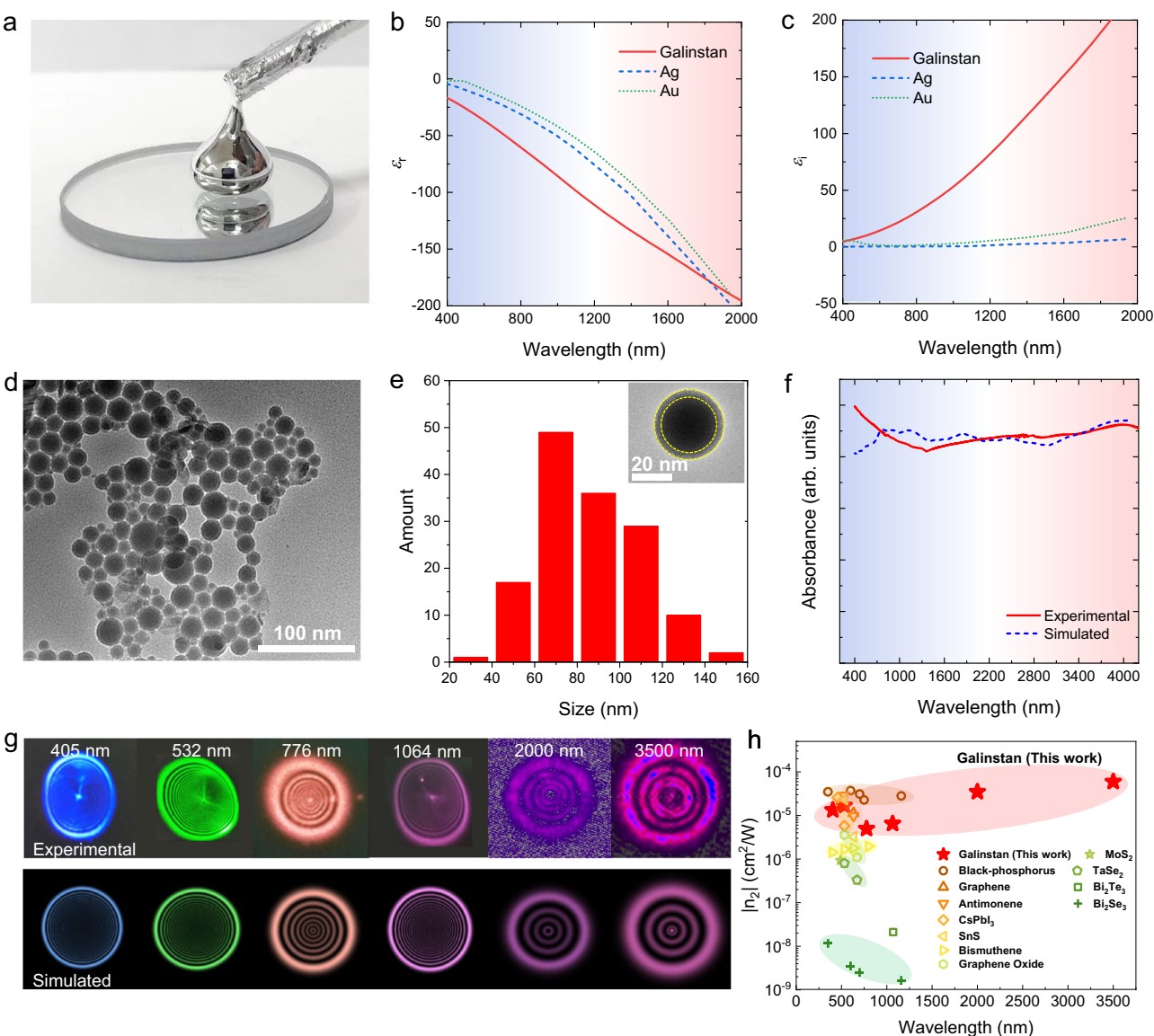

**Fig. 1 | Characterization of GNAs. a** Photograph of bulk Galinstan droplet maintaining a liquid phase well at room temperature. **b** Real and (**c**) imaginary part of the dielectric constant of Galinstan, Ag, and Au. **d** TEM of GNAs. **e** Corresponding size distribution counting of GNAs (Inset: Magnified TEM for single Galinstan nanodroplet. The dashed line notes the boundary of $Ga_2O_3$ shell). **f** Experimental and simulated visible-infrared absorption spectra of GNAs. **g** Top panels: experimental

SSPM diffraction ring patterns from the GNA dispersion for $n_2$ measurement. Bottom panels: numerical simulations based on Kirchhoff's diffraction integral. **h** Comparison of $n_2$ between GNAs and typical reported nanomaterials extracted from SSPM measurement. References: Black-phosphorus[36], Graphene[37], Antimonene[38], CsPbI$_3$[39], SnS[40], Bismuthene[41], Graphene Oxide[42], MoS$_2$[43], TaSe$_2$[44], Bi$_2$Te$_3$[45], Bi$_2$Se$_3$[46].

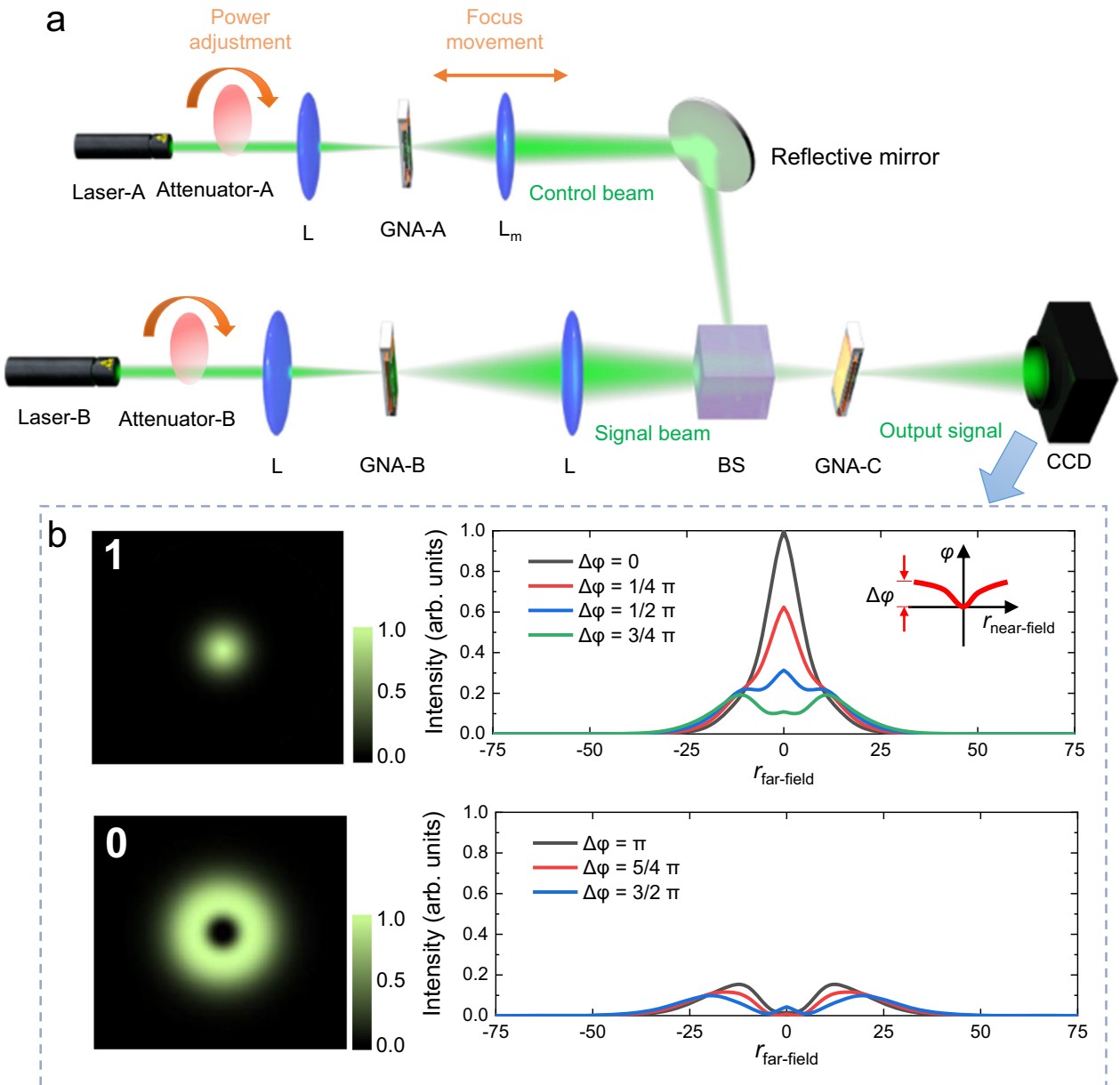

**Fig. 2 | Structure and logical states of the AOLGs based on GNAs. a** Schematic of the AOLG configuration. Attenuator: variable optical attenuator; L: focal lens; $L_m$: movable focal lens fixed on a translation platform; BS: beam splitter. **b** Dependence of '1' and '0' logical states on phase shifts. $r_{near-field}$: the radial coordinates near the focal plane. $r_{far-field}$: the radial coordinates near the CCD.

ring mode with dark far-field center as logical state '0'. Since the phase of spatial modes is important for the accuracy and stability of logical processing, the phase criteria for '1' and '0' states need to be clarified. The total phase-shift of signal beam is noted as $\Delta\varphi = \Delta\varphi_L + \Delta\varphi_{NL}$, where $\Delta\varphi_L$ is the maximum linear phase delay between the center and edge of transverse section of signal beam, and $\Delta\varphi_{NL}$ is the maximum nonlinear phase-shift of signal beam induced by the nonlinear RIF. According to Kirchhoff's diffraction formula[35], $\Delta\varphi$ determines the far-field mode as presented in Fig. 2b (see Supplementary Section IV for details). For a fundamental Gaussian beam, if the maximum phase-shift $|\Delta\varphi|$ at the exit plane of the GNAs is smaller than $\pi$ (such as around 0, $\pi/4$, $\pi/2$, and $3\pi/4$, see the top panel of Fig. 2b), the far-field center is strongly bright, namely as '1'; if $|\Delta\varphi|$ exceeds $\pi$ (such as around $\pi$, $5\pi/4$, and $3\pi/2$, see the bottom panel of Fig. 2b), the beam exhibits the first-order diffraction ring mode with a dark center, namely as '0'. It is worth noting that the control beam not only controls the signal-beam mode, but also carries logical states in virtue of the two spatial modes. The difference

between the two modes has a negligible impact on the generation and distribution of RIF, because the focus spot of control beam is much smaller than the RIF region.

In order to further clarify the TONL modulation mechanism of the AOLGs, the phase distribution of the signal-beam spatial modes responding to the dynamic control-beam RIF is carefully analyzed with the general roles depicted in Fig. 3a. Note that, for Gaussian beam, the radius of curvature of the optical wavefront is dependent on the axial distance with respect to the focal point[47], so that the distribution of transverse phase $\varphi_L(r)$ changes at different axial positions (Supplementary Fig. 8). We can thus obtain different $\Delta\varphi_L$ by controlling the distance between RIF and the focal position of signal-beam. Because the RIF has a non-uniform distribution with Gauss profile, different position of signal-beam focal plane in the RIF would produce positive $\Delta\varphi_{NL}$ (>0) of different values. In this case, we can modulate $\Delta\varphi$ conveniently through two ways: firstly, manipulating the position of RIF (namely the focus point of control beam) with respect to the focal

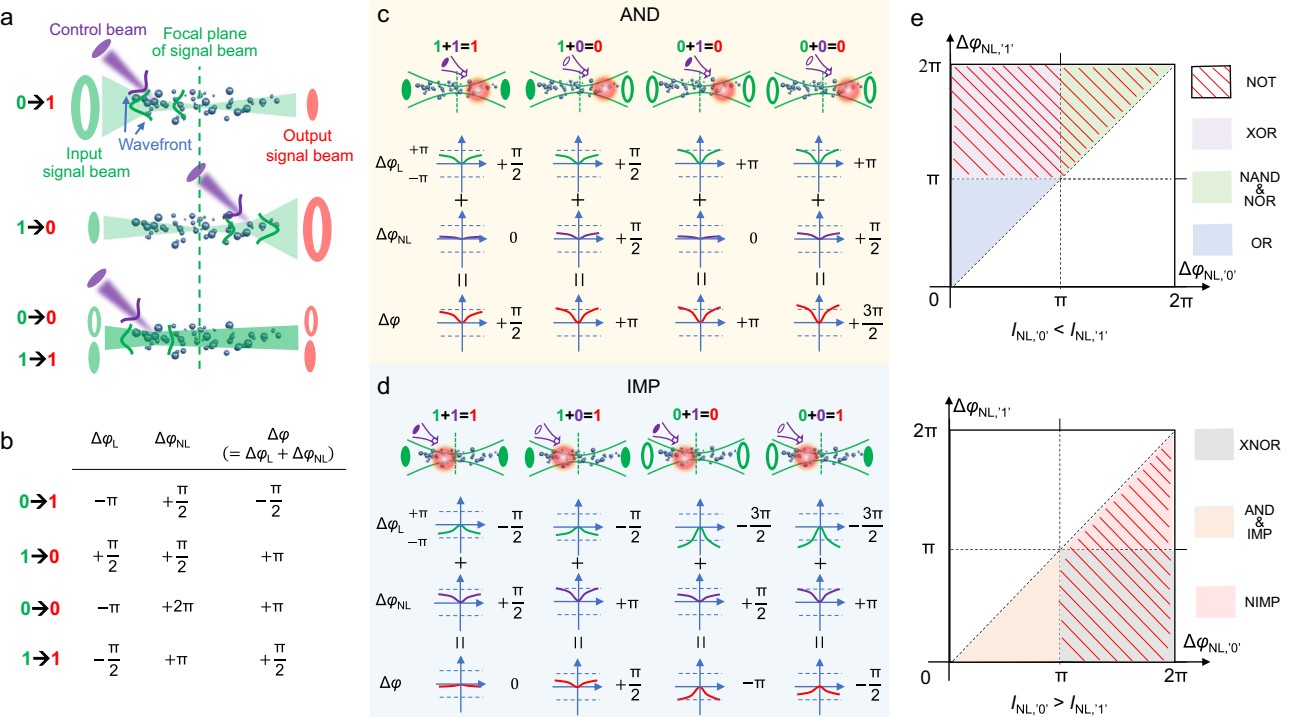

**Fig. 3 | Mechanism for reconfigurable AOLGs based on phase modulation.**
**a** Role of reversible transition between '0' and '1' via modulating the position and strength of control beam. **b** Scheme of corresponding typical phase dependence for reversible transition. Mechanism of phase modulation and corresponding scheme of phase dependence for representative (**c**) AND and (**d**) IMP logic gates. **e** Phase diagram of the control-beam strength and signal-beam nonlinear phase-shift for switching among the nine AOLGs.

plane of signal beam by moving the focal lens $L_m$ along the propagation direction of control beam; secondly, moderately turning the RIF strength by adjusting the strength of control beam via attenuator-A. Therefore, the transition between '0' and '1' can be reversible through controlling the two ways to appropriately modulate $\Delta\varphi$ (Fig. 3b), which can create unexplored opportunities for realization of versatile optical Boolean logical functions. The mechanism of phase modulation for AND and IMP gates are taken as examples to explain the AOLG design (Fig. 3c, d). For the scheme of AND, when the logical modes of input control and signal beams are both '1', the output signal is '1' with $|\Delta\varphi|<\pi$. Otherwise, the output is '0' in the case of $|\Delta\varphi|\geq\pi$. The corresponding configuration criteria of control beam strength, distance of RIF (i.e., focal position of control beam), and $\Delta\varphi$ of signal beam for implementing the phase modulation for nine logical functions are concluded in Table 1. Here the strength of control beam at '0' and '1' states of control beam is noted as $I_{NL,'0'}$ and $I_{NL,'1'}$, respectively. The correspondingly induced $\Delta\varphi_{NL}$ of signal beam to create '0' and '1' states of signal beam are indicated as $\Delta\varphi_{NL,'0'}$ and $\Delta\varphi_{NL,'1'}$, respectively.

Based on the modulation process demonstrated above, one can summarize the experimental criteria for realization of AND gate functionality. (1) The focal position of control beam is behind the focal plane of signal beam to introduce a positive linear phase of signal beam ($\Delta\varphi_L>0$). (2) $I_{NL,'0'}$ is set to be larger than $I_{NL,'1'}$ to introduce a large $\Delta\varphi_{NL,'0'}$ (around $\pi/2$) and small $\Delta\varphi_{NL,'1'}$ (around 0), which can be simply summarized as $\pi>\Delta\varphi_{NL,'0'}>\Delta\varphi_{NL,'1'}>0$. The AOLG functionality can thus be rapidly switched to IMP by moving the control-beam focus position in ahead of the signal-beam focal plane to produce $\Delta\varphi_L<0$. The IMP and its negation (NIMP) are two important gates for construction of stateful logic circuits, which can simultaneously store logical values and conduct logical operations in the same process[48]. By reassembling the configuration according to Table 1, the AOLG function is also easily reconstructable into other seven fundamental gates including OR, NOT, NOR, NAND, XNOR, XOR, and NIMP gates with respective

mechanisms described in Supplementary Figs. 9–15. Specially, NAND gate needs smaller $\Delta\varphi_{NL}$ than NOR as shown in Supplementary Figs. 11 and 12. For clearer comparison, Fig. 3e illustrates the distinct criteria of nine gates depending on the signal-beam nonlinear phase shift and control-beam strength.

## Implementation of reconfigurable all-optical logic functions

By employing the modulation criteria proposed above, we can then experimentally implement all the nine fundamental Boolean AOLG functions into one programmable circuit as depicted in Fig. 2a, with the performance of AND and IMP operation presented in Fig. 4a, b, respectively. The control and signal light sources are continuous-wave solid-state lasers with wavelengths centered at 532 nm, 1342 nm, and 2.0 μm, respectively. The left images in Fig. 4a are the experimentally obtained output patterns of signal beam for AND gate processing at a broad band crossing 532 nm to 2.0 μm with an operation bandwidth of ~1500 nm, where the focal position and strength of control beam are set based on as-mentioned criteria (the detailed conditions are given in Supplementary Table S3). These AND images indicate a high contrast of the central intensity between the states of '1' and '0' in the range of 8:1–17:1, enough for precise identification of '1' and '0' states (see Supplementary Table S2 for detail). The corresponding temporal response is presented in the right diagram of Fig. 4a, indicating a good stability with great potential for application in optical computing with higher-level cascade and integration.

The performance of function switched into IMP gate is presented in Fig. 4b (please see Supplementary Section V for more detail). Moreover, OR, NOT, NOR, NAND, XNOR, XOR, and NIMP gates can be also experimentally realized with recorded spatial and temporal performances systematically shown in Supplementary Section V and Supplementary Figs. 9–15, respectively. Accurate discrimination of the logic processing is feasible since the signal-to-noise ratios (SNRs) of the nine logic gates are higher than 4.1:1 (see Supplementary Table S2 for

**Table 1 | Criteria of the nine logic gates**

| Gate | Focal position of control beam (with respect to the focal plane of signal beam) | $\Delta\varphi_L$ | Strength contrast of control beam | $\Delta\varphi_{NL}$ |
|---|---|---|---|---|
| AND | Behind | >0 | $I_{NL,'0'}>I_{NL,'1'}$ | $\pi>\Delta\varphi_{NL,'0'}>\Delta\varphi_{NL,'1'}>0$ |
| OR | Ahead | <0 | $I_{NL,'0'}<I_{NL,'1'}$ | $\pi>\Delta\varphi_{NL,'1'}>\Delta\varphi_{NL,'0'}>0$ |
| NOT | Ahead | <0 | only $I_{NL,'0'}$ (or only $I_{NL,'1'}$) | $2\pi>\Delta\varphi_{NL,'0'}$ (or $\Delta\varphi_{NL,'1'}$)$>\pi$ |
| NAND | Ahead | <0 | $I_{NL,'0'}<I_{NL,'1'}$ | $\Delta\varphi_{NL,'1'}>\Delta\varphi_{NL,'0'}>\pi$ |
| NOR | Ahead | <0 | $I_{NL,'0'}<I_{NL,'1'}$ | $\Delta\varphi_{NL,'1'}>\Delta\varphi_{NL,'0'}>\pi$ |
| XNOR | Ahead | <0 | $I_{NL,'0'}>I_{NL,'1'}$ | $\Delta\varphi_{NL,'0'}>\pi>\Delta\varphi_{NL,'1'}>0$ |
| XOR | Ahead | <0 | $I_{NL,'0'}<I_{NL,'1'}$ | $\Delta\varphi_{NL,'1'}>\pi>\Delta\varphi_{NL,'0'}>0$ |
| IMP | Ahead | <0 | $I_{NL,'0'}>I_{NL,'1'}$ | $\pi>\Delta\varphi_{NL,'0'}>\Delta\varphi_{NL,'1'}>0$ |
| NIMP | Ahead | <0 | $I_{NL,'0'}>I_{NL,'1'}$ | $\Delta\varphi_{NL,'0'}>\Delta\varphi_{NL,'1'}>\pi$ |

$\Delta\varphi_L$: linear phase-shift of signal beam, $\Delta\varphi_{NL}$: nonlinear phase-shift of signal beam, $\Delta\varphi_{NL,'0'}$: $\Delta\varphi_{NL}$ to induce '0' state of signal beam, $\Delta\varphi_{NL,'1'}$: $\Delta\varphi_{NL}$ to induce '1' state of signal beam, $I_{NL,'0'}$: strength of control beam at '0' state of control beam, $I_{NL,'1'}$: strength of control beam at '1' state of control beam.

detail), comparable with the performance of optoelectronic logic gates based on graphene plasmons[4]. To the best of our knowledge, this reconfigurable AOLG scheme possesses the highest degree of functionality integration and largest operation bandwidth in one single configuration than the reported state-of-the-art optical logic gates (including AOLGs and optoelectronic logic gates)[1,4,7–13,49–53] as listed in Supplementary Table S4. Larger operation band from 400 to 4000 nm is expectable according to the broadband plasmon TONL of Galinstan. These advantages based on GNA plasmonic absorber would enable not only the realization of multifunctional photonic circuits, but also the implementation of massively parallel processing of high-density data in further all-optical microprocessor.

Aside from the flexible integration of multiple functionalities, the modulation speed of the proposed AOLG is promising as well, considering that recent progresses have evidenced the intriguing advancements of fast opto-thermal modulation on complex refractive index[54,55] and reversible switching of crystallographic phases[56] on picosecond timescale. As ultrafast transient absorption (TA) spectroscopy has been widely used to investigate the capability of modulation speed for a variety of optical components[51,57,58], here we carried out pump-probe TA measurement to describe the potential speed of transient modulation process in the AOLGs. As depicted in Fig. 4c, the strong pump beam will cause hot-electron transition, electron recombination, phonon scattering, and thermal diffusion in the GNA/NMP sample that modulate the intensity and spatial mode of the weak probe beam. The corresponding time-resolved process is well fitted to a bi-exponentially decaying function (Fig. 4d, e), as $\Delta T/T = y_0 + \left(a \cdot e^{-t/\tau_1} + b \cdot e^{-t/\tau_2}\right)$, implying the transfer of plasmonic energy can be divided into two steps (noted as the relaxation process 1 and 2). The relaxation process 1 is a fast process with the time constant $\tau_1$ of ~10 ps, referring to the process of carrier-carrier intraband thermalization and recombination, accompanied by phonon scattering within the GNAs. The relaxation process 2 is a relatively slow process with the time constant $\tau_2$ of ~ 200 ps, mainly arising from phonon transfer across the GNA/NMP interface and thermal diffusion in the solution for RIF generation[59]. The possible overall processing speed of the ALOGs based on entire opto-thermal process can thus be estimated as ~210 ps. This relaxation time is typically insensitive to the operation wavelength and thus exhibits broadband stability advantage (Fig. 4f). With further compression of the interaction region between the two beams in this AOLG architecture, a faster speed is expected due to shorter phonon diffusion and faster thermal equilibrium.

## Discussion

By far we have demonstrated a structured liquid metal plasmonic absorber based programmable AOLG platform with broad bandwidth and multifunctionality in spatial light configuration. Based on

numerical simulation, here we further delineate that nanoscale TONL effects of a few GNAs are capable of modulating the beam spatial mode through SXPM just as in the macroscopic system experimentally shown in this work (see Supplementary Section VI for details), suggesting the feasibility of the proposed broadband AOLG scheme in microscopic system. Therefore, apart from the proof-of-concept demonstration in the dispersion-based devices, this scheme may be applicable to nanoscale optical integration such as waveguide system. For instance, the integration of this logic scheme to all-optical circuits on waveguide platform can be expected through coating the GNAs on optical waveguides or injecting the GNAs into waveguide by ion beam, in terms of those successful realizations of various waveguide nanodevices based on metal nanoparticles[60] and two-dimensional materials[4,17]; the guided-wave propagation for both the fundamental Gaussian beam mode and first-order diffraction ring mode can be supported in a ring-core waveguide structure[61]. Quick switching among different logical functions may be achievable on integration platform by using a MEMS (microelectromechanical system)-stepping microlens/attenuator to change the focus position/intensity of control beam. Moreover, relying on the unique advantage of flexible shape morphing of GNA, flexible all-optical logic element, which is the key component for future flexible photonic circuits, is also predictable by integrating GNAs with soft photonic materials.

In summary, we experimentally demonstrate a design for a reconfigurable AOLG configuration based on the plasmon enhanced TONL enabled by the huge effective nonlinear refractive index of GNAs. This scheme has a series of unique advances, including simple all-optical configuration without external electronic modulation, all the nine fundamental Boolean logics achievable in one single configuration, ultrabroad operation band (532–2000 nm in experiment, 400–4000 nm in potential), low energy consumption, as well as easy fabrication of flexible GNAs with heterogenous plasmonic nanostructures. It is also reasonable to conclude that the strong TONL of GNAs can provide a potential avenue for exploring more novel multifunctional optical devices with miniature structures, fast optical modulation, and ultrabroad operation bandwidth.

## Methods

### Preparation of GNAs

The GNAs were synthesized by alloying and ultrasonic method. Ga (67%wt), In (20.5%wt) and Sn (12.5%wt) grains were mixed and heated at 500 °C for 30 min, and the bulk Galinstan droplet was obtained. Then 0.5 g droplet was mixed with 200 ml NMP solvent in a bottle, and processed by 180 W ultrasonication for 20 h. After standing for 24 h, the upper liquid was taken for centrifugation. The setting parameters of the centrifuge are 5000 rpm and 30 min. After centrifugation, the GNAs were obtained in the upper liquid.

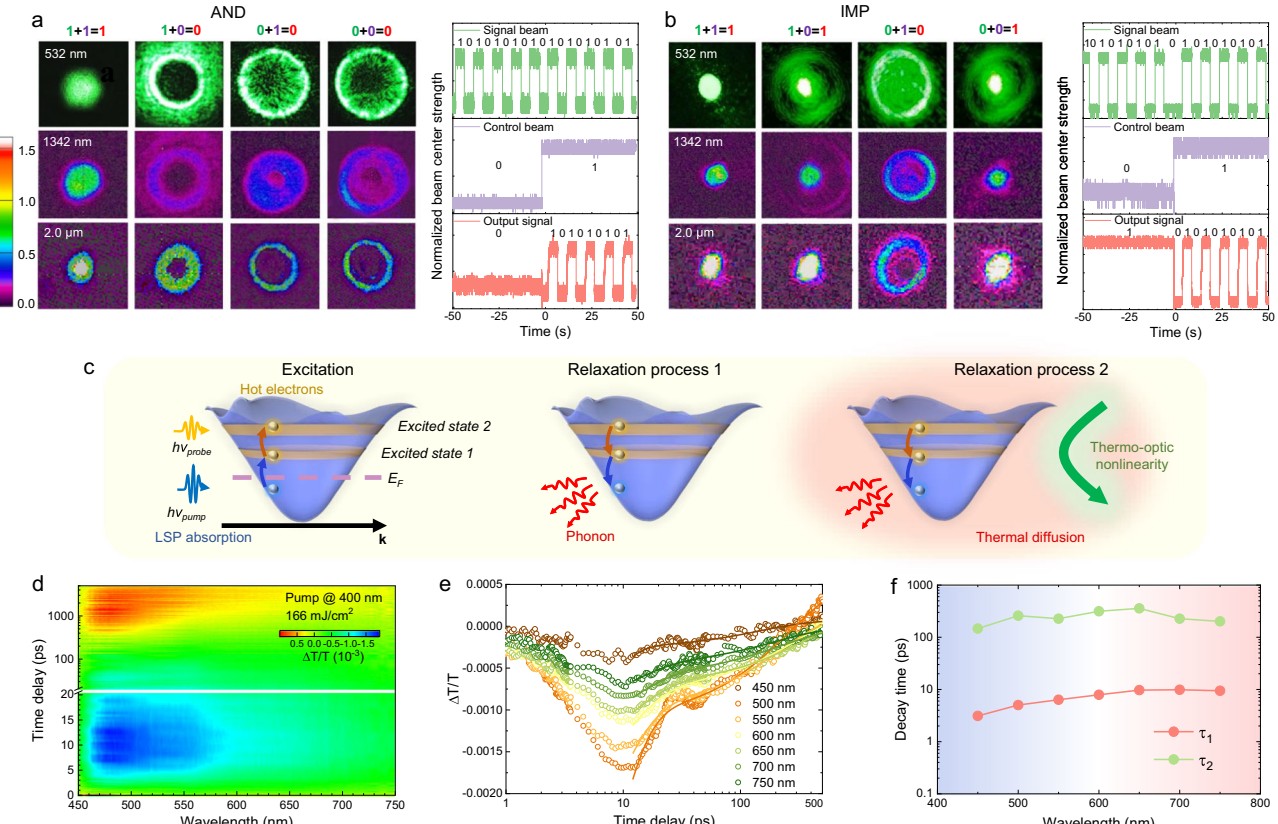

**Fig. 4 | Experimental results of reconfigurable AOLGs based on GNAs and time-dependent modulation dynamics. a** Left: spatial output patterns of AND gate across visible and infrared region. Right: temporal input and output in AND gate processing. **b** Left: spatial output patterns of IMP gate with broadband response. Right: temporal input and output in IMP gate processing. **c** Schematic mechanism for the decay states. **d** Time-resolved absorption spectrum measurement. **e** Time-resolved differential transmission at different excitation wavelengths. Solid line: bi-exponential model fitting. **f** Corresponding two decay times after hot-electron generation as a function of different probe wavelengths.

## SSPM experiments with GNAs

The GNAs were sealed into a 1-mm-thick cuvette (JGS1 quartz) with an optical length of ~1 mm. All lasers (405, 532, 776, and 1064 nm) were focused vertically into the Galinstan/NMP dispersion from a top convex lens. Due to SSPM effect, the transmitted beam was diverged into several rings. The number of rings was counted, and the incident strength was recorded simultaneously. Vertical incidence was applied to avoid the collapse of diffraction rings during non-axis-symmetric thermal convection.

## All-optical logic gate experiments with GNAs

The three GNA samples sealed in 1-mm-thick cuvettes (JGS1 quartz) were cascaded with GNA-A and GNA-B as signal generators and GNA-C as processor. A series of home-made and commercial continuous-wave solid-state lasers with wavelengths centered at 532 nm, 1342 nm, and 2.0 μm were employed as the light source Laser-A and Laser-B. The strength of two beams was dynamically controlled by variable optical attenuator. The two lasers were focused into GNA-A and GNA-B to produce the input '0' or '1' signals, followed by simultaneously focused into GNA-C to implement logic operation. The output signal beam was detected by the CCD camera. Importantly, the focal position of control beam relative to focal plane of signal beam and the strength of control beam were specifically controlled for different logic operations. The light transmission loss of the 1-mm-thick GNA/NMP sample was measured to be 13–15% (the total of reflection loss and absorption) within the wavelength region of 532–2000 nm, corresponding to an insertion loss of 0.6–0.7 dB. All the experiments were carried out at room temperature.

## Transient absorption measurement

TA experiment was conducted using a commercial Ti:Sapphire regenerative amplifier (Libra, Coherent) at 800 nm with a repetition rate of 1 kHz and pulse duration of ~90 fs. An optical amplifier (OperA solo, Coherent) pumped by the regenerative amplifier was used to provide a pump beam with tunable wavelength. The probe supercontinuum source covering the spectral range from ultraviolet to near-infrared was generated by focusing a small portion of the femtosecond ultrashort pulses on a 5 mm CaF$_2$ plate. The TA signal was then analyzed by a high-speed charge-coupled device (S7030-1006, Hamamatsu) with a monochromator (Acton 2358, Princeton Instrument) at 1 kHz enabled by a custom-built control board (Entwicklungsbüro Stresing).

## Data availability

The authors declare that the data supporting the findings of this study are available within the main text and Supplementary Information, including in the "Methods" section and Source Data files. Source data are provided with this paper.

## Code availability

The date and code used in this study are available on Zenodo under the accession code: https://doi.org/10.5281/zenodo.10469643.

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

## Acknowledgements

We thank Prof. Chunfeng Zhang in Nanjing University for assistance with the TA measurements and Prof. Jimin Zhao in Institute of Physics, Chinese Academy of Sciences for assistance with the numerical calculations. The authors acknowledge financial support from the National Key R&D Program of China (Grant Nos. 2022YFA1404300 to L.Z., 2019YFA0705004 to Z.D.X.), the National Natural Science Foundation of China (Grant Nos. 92150302 to J.L.X. and X.P.H., 12022403 to L.Z., 12334015 to J.L.X., 62293523 to Z.D.X., 62375122 to J.L.X., 12174185 to X.P.H., 92163216 to X.P.H.), Guangdong Major Project of Basic and Applied Basic Research (Grant No. 2020B0301030009 to Z.D.X.), Zhangjiang Laboratory (Grant No. ZJSP21A001 to Z.D.X.).

## Author contributions

L.Z., J.L.X., Z.D.X. and X.P.H. conceived and supervised the project. J.L.X., C.Z. and M.D.W. designed the experiments. M.D.W. and Y.M.X. prepared the Galinstan samples and performed material morphology characterization. J.L.X., C.Z., Y.L.W. and M.D.W. performed the optical measurements and analyzed the data. C.Z., Y.L.W., T.Q.W. and S.Q.L. performed the numerical simulations. J.L.X., C.Z., Y.L.W. and L.Z. wrote the manuscript with the assistance from Z.D.X., X.P.H. and S.N.Z. C.K.L., G.Z., X.J.L., H.Z. and S.N.Z. contributed to the discussions.

## Competing interests

The authors declare no competing interests.
