## [Peer Review File · Nature Communications]

All-in-one, all-optical logic gates using liquid metal plasmon nonlinearityReviewer #1 (Remarks to the Author):

The manuscript of Xu and co-workers report a scheme to realize ultra-broadband all-in-one all-optical logic gates, using the liquid metal plasmon nonlinearities. They show that refractive index of liquid-metal Galinstan nanodroplets can be adjusted by controlling the strength and focal position of control beam. This scheme brings multiple variation of signal-beam phase shift. The author successfully constructed 9 basic logic gates in a circuit configuration. The photonic control on plasmon enhanced thermo-optical nonlinearity is very interesting, and the paper is well organized. I can recommend to publish this work in Nature Communications, after the authors well addressed the following questions.

1. In the introduction, the authors claimed "we for the first time demonstrate the reconfigurable all-in-one AOLGs with operation bandwidth ranging from visible to infrared range". This is unfair. Actually scientists have realized nonlinear plasmon based optical logic gates covering the NIR-MIR band [Ref. Nature Communications 13, 3138 (2022): 10.1038/s41467-022-30901-8], based on the gate tunable plasmonic frequencies [Ref. Nature Photonics 12, 22 (2018): 10.1038/s41566-017-0054-7]. Moreover, the aforementioned scheme is based on photon-electron interaction, much faster than the opto-thermal effect. Can the authors discuss their unique advances more accurately?
2. The authors claimed that "due to the diffraction limitation of each optical component, it is almost impractical for large scale optical integration by simply packaging different logic gates on a chip as widely employed in electric circuits." I agree. But in this work, plasmonic generation relies on lens group based free space focusing and a CCD (e.g. Fig. 2), such an operation cannot be called 'integrated'. I suggest the authors turn down their voice.
3. The authors claimed that the nonlinearity is enhanced by plasmons. However, there is no quantitative description about the plasmons all through the paper. What is the SPR frequency? How can the plasmons be confined (or, did the authors know the wavelengths of the plasmons when the exciting the liquid metal drops by using different optical frequencies)? Can the author plot the dispersion curve of the plasmons?
4. According to Fig. 1d, there are many GNAs in a nanoscale window. 1) It is for sure that the light spot size is much larger than a GNA. Therefore, multiple GNAs are simultaneously excited. Would the light beams scattering from different GNAs interfere with each other? 2) Since the GNAs are closed to each other spatially, for a typical metal plasmon wavelength in tens of nanometer level, would the signals coupling with each other? 3) Is the 'concentration' of the GNAs important, can the authors clarify if they can control the GNAs' concentration, can they control the size of a GNA?
5. The authors realize logical operations by controlling $\Delta\phi$. Can the authors provide the relationship between the control beam intensity, the distance of the RIF, and the focal plane of signal beam on $\Delta\phi$?
6. The authors pay too much attention to the conceptual description of the all-optical logic gate in the text, and lacks a description of the specific test conditions for realizing the functions, such as the experimental environment, device parameters, the optical power and distance between the RIF and the focal point of the signal beam. It is recommended that the author supplement the experimental conditions more in details.
7. Since the GNAs demonstrate strong absorption, even as high as 0.8 (SI FIG 4). That is why they must use pulsed lasers. What is the power consumption? Can the authors discuss the related limitations?
8. Scale bars in all the colored maps are missing. Fonts in Fig. 4 are too small.
9. The title is a bit too long to read, and seems technical. I strongly suggest the authors can consider this minor point.

Reviewer #2 (Remarks to the Author):

Because of their fast response, all optical logic gates (AOLGs) are of great significance in large-scale integrated optical processors. This work experimentally demonstrates the ultrabroadband AOLGs enabled by reconfigurable plasmon enhanced thermo-optical nonlinearity (TONL) of liquid-metal Galinstan nanodroplet assemblies (GNAs). This is an interesting work with sufficient evidence, and I

would love to see it published after minor revisions.

1. The intensity at the center of a Gaussian beam is usually much stronger than the edge, so it is hard to achieve perfect destructive interference. One example is the first line of Fig. 1g. Please comment on this issue.
2. Some of the references of logic gate and refractive index in supplementary information (SI) should be included in the main text, especially those related to plasmonic method and GNAs.
3. Is it the same to use Au Nps with different size? Also, the solid-state Au Nps might be better for integration than the liquid-state GNAs. By the way the discussion in SI about possible integration of GNAs is not sufficient.
4. In SI it was said: Constructive interference occurs when the phase difference equals to $(2m-1)(\pi/\lambda)$, with the appearance of bright ring referred to as mth-order diffraction ring mode. Is it correct?
5. In SI Figure 8 and subsequent figures, the sequence in (a) and (b) is not consistent: 0+0, 0+1, etc.
6. For all the logic gates, please give the contrast between state 1 and state 0. Also, please give the SNR.

Reviewer #3 (Remarks to the Author):

The authors demonstrate a reprogrammable/multifunctional optical logic gate (from 532-2000 nm) based on spatial cross phase modulation using liquid-metal Galinstan nanodroplets. The nanodroplets show a novel approach to engineering/exploiting high nonlinearity required for cross-phase modulation through plasmon enhanced photothermal effects. By simply tuning the power and placement of the beam waist of a probe beam at respect to the beam waist of a signal beam (in which the nanodroplets are immersed), the authors control the nonlinear phase of the output cross phase modulated beam, resulting a desired higher-order mode Bessel diffraction pattern. A mode with a bright center gives a Boolean "1" and higher order modes with a dark center give a "0." In a single setup, the authors are able to show nine different logic gates through tuning control beam power and placement.

The experiments are thorough and well done, including careful characterization of the nanodroplets and their complex nonlinear susceptibility. As far as I can tell, the science is solid and sound. Figures are well designed and beautiful (really nice!). I also appreciate the long supplement that gives detailed background on the work.

Though the work is clever and executed well, I unfortunately don't think it merits publication in a journal like Nature Communications because it is too incremental (Note that most scientific work is incremental; that's not a bad word! That's how we move forward.). I don't think this work would be attractive to a broad audience or have a broad impact. That's o.k. That just means it fits somewhere else- a more technical, field specific journal. I give more detailed evidence for this below. Because some of this is opinion, I also comment on revisions that should be addressed if the editors and other reviewers wish to see the work published in Nature Communications or as helpful criticism for the authors to use as they revise and try to publish their work elsewhere.

Issues with Broader Impact of Work:

The field of optical signal processing has rightly come under intense scrutiny over the last decade due to practical bottlenecks in information and communication technology, for which simply switching them to optical systems won't address. We know that the major cost and problems with the speed of information is energy consumption and heat dissipation. John Hennessy's 2018 talk "The Future of Computing" at google I/O gives a very nice explanation for this. However, researchers like David A. B. Miller have been discussing this as early as 2010, as noted in his well cited commentary article in Nature Photonics "Are Optical Transistors the Logical Next Step." Rodney Tucker and Kerry Hinton give a detailed assessment of the problem of energy consumption in their 2011 review, "Energy Consumption and Energy Density in Optical and Electrical Signal Processing." The short story is that current commercial CMOS electronics is much more energy efficient than any optical system shown to

date. Electronic transistors consume ~ 0.1 fJ per bit. At speeds of ~ 10 GHz, this amounts to a microwatt of average power.

Optics, especially when nonlinearity is employed is energy greedy. The best nonlinear optical gates to date are at the ~ 1 pJ per bit level, 10,000x worse for energy than current commercial CMOS. Optical gates using linear interference have shown much less energy consumption, but have other issues- integration, fan-out, scalability, miniaturization. The moral of the story is that optical signal processing for optical processing sake is really not broadly interesting anymore. And certainly, simply switching from electronic systems to optical is not the way to speed up information- which unfortunately is an argument that many optical scientists still make.

The use of optical processing is of great importance still, but it's more nuanced: as it relates to networking and transmission, interconnects, and most definitely quantum information (likely the future of computation and communication).

The authors first statement in the abstract is,

"Multifunctional all-optical logic gates (AOLGs) of massively parallel processing are of great importance for large-scale integrated optical processors with executing speed far in excess of electronics, while are rather challenging due to limited operation bandwidth and multifunctional integration complexity."

That's fair. They are being honest. However, this implies that large-scale integrated optical processors are interesting in their own right. This work therefore may or may not (it's research after all) help people mainly in the field of optical processing.

If this work could be applied more broadly to information processing, solve an issue with energy consumption for high-speed signals, applied to quantum information processing, or nanoscale optical integration, this would have a broader impact and warrant publication in a scientific journal with readership of broad scientific backgrounds.

Suggestions for Revision:

1. If the editor disagrees with my assessment of lack of broad impact and decides to publish this work in Nat. Comm, the motivation (abstract and introduction) needs to start much broader. It should be laid out as

-Why high-speed logic is interesting to information processing and communications broadly (not just optics or plasmonics)

-What are bottlenecks in high-speed logic (not just plasmonics)

-What have others done to solve this problem (not just nonlinear optics)

-What are the shortfalls they are not meeting: lack of single platform multifunctional device, small wavelength operating range, small scale integration, etc.

-Then how the technique using plasmon enhanced liquid metal nanodroplets solves it

2. In the beginning of the discussion section the authors claim that the technique could be scaled to the micro-optic, on-chip level. The authors give some proposed details: coating waveguides with GNAs, ion beam injection, using MEMs mirrors to steer beams, etc. They give some simulation results of their technique at small scale, however, not in a prototype waveguide system.

The simulation in the supplement therefore is too simple/superficial at this point to be included and there is a lot of guessing what this system might look like. The authors should remove their simulation results until they have a simulation in an actual waveguide structure like what they show in S15 d) as well as dial down scaling claims. The simulation is a good start, just not compelling enough now.

The cuvette the authors use in their proof of concept is 1 mm thick. That's pretty macroscopic. Would just a few GNAs actually work in a scaled down version as they say? What is the optical thickness of GNAs used in the simulation? This is not mentioned and an important detail.

Note: it may be fair to make some gentle/softer claims about scaling down to chip level- 1-3 sentences with references. Or it could be o.k to include the simulation IF much, much more details of the simulation parameters are given. Regardless, the schematic of the proposed circuit, S15 d should be removed. This schematic is a nice idea but not tested at all by the authors' simulation or backed up by reference.

3. Order is a little mixed in my opinion in the first few sections. I would suggest explaining the concept of the device generally first (beam with bright center/beam with dark center) then give background on SXPM and the nanodroplets. In fact, the section characterizing the nanodroplets really should be moved to the supplement. As it is, it is confusing to have a lot of details about nanodroplets at the beginning of the paper when the main point of the work is about broad wavelength operation multi-function optical logic.

Alternatively, another approach would be to give the section explaining measurements on characterizing the nanodroplets (the section with Fig 1) a different name, like "Characterization of Nanodroplets." This way the reader would know that this is specialized information about part of the system.

4. References:

The statement,

"The past decade has witnessed vast advancements AOLGs through linear or nonlinear optical modulation of nano-semiconductors, ranging from stimulated scattering and photoluminescence (PL) of nanowires or nanospheres¹⁰⁻¹² to spatial self-phase modulation (SSPM) or spatial cross-phase modulation (SXPM) of nanosheets¹³⁻¹⁵"

is very under-referenced. More should be included. I would recommend the authors look at two of the references they already cite in the main paper and the supplement:

1) Minzioni, P. et al. Roadmap on all-optical processing. J. Opt. 21, 063001 (2019)

2) Maram, R. et al. Frequency-domain ultrafast passive logic: NOT and XNOR gates. Nat. Commun. 11, 5839 (2020).

The authors may be able to use some of the references in these works or forward cite appropriate references from these papers for current work across AOLGs. Both have a lot of review information on the field.

Note, in general, that optical logic gates fall into two main categories of operation principle: 1) nonlinearity and 2) interference. The list the authors cite in this section, refs 10-15 are really all nonlinear devices. This needs to be changed (i.e. add at least one reference that actually uses linear interference)

Note also that Maram et al's work "Frequency-domain ultrafast passive logic" is expressly not a nonlinear technique- hence the word "passive" in the title. The authors incorrectly label this work as using "nonlinear fiber" in the chart in the supplement and so this should be changed. This work is an example of linear interference.

4. The English needs to be tidied up a bit. It's mostly fine and consists mainly of a handful of small errors such as subject verb agreement (easily corrected in copyediting). However, there are some statements that were confusing to me or are particularly important to edit.

Title: "Liquid-metal plasmon nonlinearity lights large-bandwidth all-in-one all-optical logic gates"

I don't understand the use of the word, "lights" here.

Perhaps change the title to, "All-in-one, all-optical logic gates using liquid metal plasmon nonlinearity"

Abstract: change "spital" to "spatial"

Introduction: I don't understand the following statement,

"In addition, due to the diffraction limitation of each optical component, it is almost impractical for large scale optical integration by simply packaging different logic gates on a chip as widely employed in electric circuits. Otherwise, complex integration of optical processors by diverse AOLG elements is bulky and high energy-consuming. Therefore, extra rational strategies for compact multifunction-integrated and broadband optical logic operators have been urgently sought."

What is it about diffraction that limits each component? How does bulk lead to energy consumption?

Also, there are a handful of awkward modifiers, "extra rational" "almost impractical," "urgently sought." These phrases should be changed.

This statement seems to be saying that different optical logic operators will need to be made physically different. This will give different spatial modes or entail many components having different spatial mode requirements, now making each component bulky, unique and not easily printable in a universal way like electronics.

Is that what this is trying to say?

Body: The word phrase "preciously investigate" in line 252 is a bit bizarre. Just remove the word "precious."

Responses to Reviewer #1

Overall Comments: *“The manuscript of Xu and co-workers report a scheme to realize ultra-broadband all-in-one all-optical logic gates, using the liquid metal plasmon nonlinearities. They show that refractive index of liquid-metal Galinstan nanodroplets can be adjusted by controlling the strength and focal position of control beam. This scheme brings multiple variation of signal-beam phase shift. The author successfully constructed 9 basic logic gates in a circuit configuration. The photonic control on plasmon enhanced thermo-optical nonlinearity is very interesting, and the paper is well organized. I can recommend to publish this work in Nature Communications, after the authors well addressed the following questions.”*

Our Reply:

We thank the Reviewer for his or her constructive efforts on evaluating our manuscript with precise summary and positive comments. Based on these comments and/or suggestions, we have carefully revised the manuscript accordingly to improve the quality. The more detailed point-to-point responses are as follows. The corresponding changes to the original manuscript are marked in purple in the responses and in red in the revised main text and supplementary.

Comment 1: *“In the introduction, the authors claimed “we for the first time demonstrate the reconfigurable all-in-one AOLGs with operation bandwidth ranging from visible to infrared range”. This is unfair. Actually scientists have realized nonlinear plasmon based optical logic gates covering the NIR-MIR band [Ref. Nature Communications 13, 3138 (2022): 10.1038/s41467-022-30901-8], based on the gate tunable plasmonic frequencies [Ref. Nature Photonics 12, 22 (2018): 10.1038/s41566-017-0054-7]. Moreover, the aforementioned scheme is based on photon-electron interaction, much faster than the opto-thermal effect. Can the authors discuss their unique advances more accurately?”*

Our Reply:

We thank the Reviewer for his or her valuable suggestions about discussions on unique advances of the proposed AOLG scheme. Great efforts for providing important references and careful proof-reading on the descriptions are greatly appreciated as well.

(1) The unique advances of our proposed AOLG scheme.

Compared with most optical logical gate schemes reported thus far, the proposed all-in-one AOLG based on the strong TONL of GNAs exhibits at least three aspects of

unique advances as follows:

(i) It is the first experimental work of nonlinear plasmon based optical logic gate scheme with **all the nine individual Boolean logic units enabled in a single configuration**;

(ii) It experimentally expands the operation wavelength from 532 nm to 2000 nm, which to the best of our knowledge **represents the largest operation bandwidth** (~1500 nm in experiment and ~3600 nm in theory based on the enhanced nonlinearity ranging from 400 – 4000 nm) for optical logic gates reported thus far;

(iii) Our AOLGs are based on the **continuous-wave dual-beam nonlinear all-optical configuration**, which can distinctly **reduce both the power consumption** (with respect to the widely used pulse laser counterparts) **and the footprint** (with respect to the linear interference scheme);

Apart from the above three unique advances, our proposed AOLG scheme will be extremely competitive for the **highly scalable lithography-free fabrication, indicating high operation tolerance without optoelectrical delay** compared to the optoelectrical schemes. Because of these advances, this AOLG strategy should be a competitive candidate for further development of all-optical processor towards miniaturization, bandwidth, scalability as well as multifunctionality.

Corresponding revisions:

In order to make the unique advances of the proposed reconfigurable all-in-one AOLG clearer, the statements in the revised manuscript have been modified as follows.

Line 9 in Page 4: “In this work, we exploit the plasmon-enhanced TONL of liquid-metal GNAs to demonstrate a reconfigurable all-in-one AOLG based on dynamic and programmable manipulation on the reversible phase shift of dual-beam SXPM interaction. In such a single all-optical configuration without external electronic modulation, nine fundamental Boolean logics are achievable including AND, OR, NOT, NOR, NAND, XNOR, XOR, material implication (IMP), and not material implication (NIMP). The ultrabroadband (400 - 4000 nm) plasmon-enhanced light harvesting of GNAs endows a logic operation band ranging from visible to infrared regions.”

Line 3 in Page 15: “In summary, we experimentally demonstrate a design for a reconfigurable AOLG configuration based on the plasmon enhanced TONL enabled by the huge effective nonlinear refractive index of GNAs. This scheme has a series of unique advances, including simple all-optical configuration without external electronic modulation, all the nine fundamental Boolean logics are achievable c in one single configuration, ultrabroad operation band (532 to 2000 nm in experiment, 400 to 4000 nm in potential), low energy consumption, as well as easy fabrication of flexible GNAs with heterogenous plasmonic nanostructures.”

(2) Clarification on the novelty statements “we for the first time ...”.

There are probably misunderstandings in the sentence “we for the first time demonstrate the reconfigurable all-in-one AOLGs with operation bandwidth ranging from visible to infrared range.” We do not only just refer to the property of ultrabroadband processing, but also mean that it is the first report on the reconfigurable all-in-one AOLG that achieves all the nine fundamental Boolean logic functions (AND, OR, NOT, NOR, NAND, XNOR, XOR, IMP, and NIMP) in one single configuration, the operation bandwidth of which can in theory be pushed up to 360 nm across visible to the mid infrared range (400 - 4000 nm).

Corresponding revisions:

To avoid the misunderstanding, the statement in the revised Abstract and introduction have been revised as follows (**Line 7 in Page 2**).

“Here we for the first time experimentally demonstrate the reconfigurable all-in-one broadband AOLG that achieves nine fundamental Boolean logics in a single configuration, enabled by ultrabroadband (400 - 4000 nm) plasmon-enhanced thermo-optical nonlinearity (TONL) of liquid-metal Galinstan nanodroplet assemblies (GNAs).”

(3) Discussions on the time response property of opto-thermal modulation.

Opto-electric modulation is commonly regarded to be much faster responsive than the opto-thermal effect based counterparts. However, recent progresses have evidenced the intriguing advancements in fast responsive opto-thermal based modulation, reaching the timescale of picosecond. For example, the opto-thermal effect can be employed to implement efficient modulation of refractive index in semiconductor within several picosecond scale [*Appl. Phys. Lett.* 118, 211105 (2021); *Nanophotonics* 11, 4073 (2022)] and activate the reversible switching of crystallographic phases in phase-change materials with the speed from several picoseconds to hundreds of picoseconds [*Nature* 336, 1566 (2012); *Adv. Opt. Mater.* 6, 1800257 (2018); *Adv. Mater.* 31, 1808157 (2019); *Nano Lett.* 20, 4638 (2020)].

In our work, we also carried out pump-probe TA measurement to describe the potential speed of transient modulation process in the AOLGs as shown in **Fig. R1**. The possible overall processing speed of the AOLGs based on the entire opto-thermal process can be estimated as ~ 210 ps. This relaxation time is typically insensitive to the operation wavelength and thus exhibits broadband stability advantage (**Fig. R1**). With further compression of the interaction region between the two beams in this AOLG architecture, a faster speed is expected due to shorter phonon diffusion and faster thermal equilibrium.

Fig. R1 (Fig. 4d-f) a, Time-resolved absorption spectrum measurement. b, Time-resolved differential transmission at different excitation wavelengths. c, Corresponding decay time after hot-electron generation as a function of different probe wavelengths. The data are fitted by bi-exponential model.

Corresponding revisions:

In order to make the unique advances of the proposed reconfigurable all-in-one AOLG clearer and avoid the misunderstanding, the statements have been modified as follows in the revised manuscript.

Line 7 in Page 13: “Aside from the flexible integration of multiple functionalities, the modulation speed of the proposed AOLG is promising as well, considering that recent progresses have evidenced the intriguing advancements of fast opto-thermal modulation on complex refractive index^{53,54} and reversible switching of crystallographic phases⁵⁵ on picosecond timescale.”

In addition, the two prestigious references have been added in the revised reference list.

[4] Li, Y. et al. Nonlinear co-generation of graphene plasmons for optoelectronic logic operations. *Nat. Commun.* **13**, 3138 (2022).

[18] Yao, B. et al. Broadband gate-tunable terahertz plasmons in graphene heterostructures. *Nat. Photon.* **12**, 22-28 (2018).

Comment 2: “The authors claimed that “due to the diffraction limitation of each optical component, it is almost impractical for large scale optical integration by simply packaging different logic gates on a chip as widely employed in electric circuits.” I agree. But in this work, plasmonic generation relies on lens group based free space focusing and a CCD (e.g. Fig. 2), such an operation cannot be called ‘integrated’. I suggest the authors turn down their voice.”

Our Reply:

We thank the Reviewer for pointing out this issue, which may originate from misunderstandings on the definition of “integrated”.

As demonstrated in Reply 1, the unique advances (with direct experimental

demonstration) are the “ultrabroadband” and “multifunctional” instead of “integrated”. That is why neither the title nor the abstract includes any “integrated” related expressions, except for the last sentence “Our work will provide a powerful strategy for next-generation broadband all-optical integrated circuits” in the abstract as the discussion on the future prospects.

We do have sometimes employed the word “integrated” in the original manuscript to describe the proposed AOLG devices mainly because of two aspects of considerations. On the one hand, the experimentally explored lens-group-based AOLG can provide the nine-in-one functionality integration in one single configuration, superior to most of their counterparts. On the other hand, the nanoscale AOLG scheme directly inspired by the lens-group-based AOLG provides the possibility of the point-of-use spatial integration.

Corresponding revisions:

In order to avoid the misunderstanding on the definition of “integrated”, we have carefully modified the corresponding expressions in the revised manuscript.

- (1) **Line 16 in Page 2:** In abstract, the sentence “Our work will provide a powerful strategy for next-generation broadband **all-optical integrated circuits**” has been revised as “Our work will provide a powerful strategy on **large-bandwidth all-optical circuits for high-density data processing in the future.**”
- (2) **Line 28 in Page 3:** “Therefore, extra rational strategies for compact **multifunction-integrated** and broadband optical logic operators have been urgently sought” has been revised as “In this case, more rational strategies have been long pursued for the further development **towards high scalability and multifunctionality.**”
- (3) **Line 26 in Page 12:** “These advantages based on GNA plasmonic absorber would enable not only the realization of **high-integrated photonic circuits...**” has been revised as “These advantages based on GNA plasmonic absorber would enable not only the realization of **multifunctional photonic circuits...**”
- (4) **Line 14 in Page 14:** “Here we demonstrate the AOLG scheme compress to the nanoscale **based on single or a few GNAs inserted waveguide systems** in numerical simulation (Supplementary Section VI), exploring **the feasibility of the waveguide or flexible integration platform.**” has been revised as “Based on numerical simulation, here we further delineate that **nanoscale TONL effects of a few GNAs** are capable of modulating the beam spatial mode through SXPM just as in the macroscopic system experimentally shown in this work (see Supplementary Section VI for details), suggesting **the feasibility of the proposed AOLG scheme in microscopic system.**”

- (5) **Line 18 in Page 14:** “Apart from the proof-of-concept demonstration in the solution-based devices, we suggest that the proposed broadband AOLG route **can be flexibly extended to the on-chip platforms, which are crucial for next generation high density optical integration.**” has been revised as “Therefore, apart from the proof-of-concept demonstration in the dispersion-based devices, **this scheme may be applicable to nanoscale optical integration such as waveguide system.**”

Comment 3: “*The authors claimed that the nonlinearity is enhanced by plasmons. However, there is no quantitative description about the plasmons all through the paper. (1) What is the SPR frequency? (2) How can the plasmons be confined (or, did the authors know the wavelengths of the plasmons when the exciting the liquid metal drops by using different optical frequencies)? (3) Can the author plot the dispersion curve of the plasmons?*”

Our Reply:

Thanks for the Reviewer to pointing out the issues on quantitative details of the particle plasmons of liquid metal droplets. Basically, there are two types of surface plasmon modes that can enable nonlinearity enhancement, i.e., the propagating SPP (widely employed in the flat plasmonic materials like graphene or metasurfaces) and the localized or particle SPR (employed in this work). It is worth noting that the optical properties (including the excitations, mode dispersion, and field confinement) of the particle SPR are quite different from the SPP (see the explanations in the item (3) below for details).

The ultra-broadband thermo-optical nonlinear response is directly related to the unique microstructure of the self-assembled close-packed stretchable liquid metal nanodroplets (or nanoparticles). The underlying physical mechanism is the plasmon hybridization induced bandwidth expansion as well as the nanometer-scale electromagnetic field concentration nearby the curved metallic surfaces, the validity of which has been successfully evidenced in our previous works in three-dimensional metallic nanoparticle assemblies [*Sci. Adv.* 2, e501227 (2016), *Nat. Photon.* 10, 393 (2016)].

(1) The LSP frequency of the liquid metal nanodroplets

The LSP frequency of an isolated plasmonic nanoparticle is defined by the collective oscillation of free electrons nearby the curved metal surface of the nanoparticle, which can be easily obtained by Mie theory (analytically) or by FDTD simulation (numerically). Unlike conventional noble metal nanoparticles, the LSP of the Galinstan nanodroplet assemblies (GNAs) is intensively hybridized because of the unique advantages of GNAs such as wide-distribution particle size, excellent inherent

stretchability of liquid phase, and strong resonant interparticle coupling, etc [*J. Phys. Chem. B* 107, 668-677 (2003), *J. Chem. Phys.* 116, 6755-6759 (2002), *J. Appl. Phys.* 98, 011101 (2005)]. Their combination provides a high density and numerous kinds of hybrid plasmonic modes, enabling efficient bandwidth broadening with respect to isolated nanoparticles. **Fig. R2** shows the representative microstructures of the nanodroplets assemblies, the simulated spectral broadening effect of the absorption cross sections of finite nanodroplets, as well as the measured and simulated absorption spectra of a macroscopic liquid metal sample, respectively. One may clearly find that the broad operation bandwidth nature of the plasmon enhanced nonlinearity effect can be mainly ascribed to the unique particle hybridization and structure morphing of the liquid metal droplets.

Fig. R2. Morphology characterization and broadband absorption of GNAs. **a** (**Fig. 1d**), TEM of GNAs. **b** (**Supplementary Figure 3e**), SEM of GNAs morphed into massive irregular geometries. **c** (**Supplementary Figure 5a**), FDTD simulation of LSP absorption spectra for six typical situations of single or few Galinstan nanodroplets with different stretchability and interparticle coupling. **d** (**Fig. 1f**), Experimental and simulated visible-infrared absorption spectra of GNAs.

Based on the above analysis, one may arrive the conclusion that the isolated nanodroplets exhibit broad heterogeneity and thus the corresponding LSP resonance frequency of nanodroplet assemblies can be in a wide range of wavelength range. In

other words, it is hard and unnecessary to identify each LSP wavelength for each droplet.

Corresponding revisions:

To show the LSP resonances depending on the stretchability and interparticle coupling of GNAs, **Supplementary Figure 5** and related demonstration have been added in supplementary S. II.

(2) LSP field confinement of flexible GNAs

As mentioned above, the high density of plasmonic modes stemmed from the wide-distribution sizes and flexible stretchability of GNAs leads to strong confinement of LSP field with an ultrabroad band across visual to infrared regions. This effect can be further verified by simulating the LSP field of randomly distributed GNAs as delineated in **Fig. R3a**. One can see that many hot spots of LSP with strong field confinement can generate stably under wideband excitation. The change of hot-spot position with excitation wavelength among different gaps of adjacent GNAs evidences the hybridization of different plasmonic modes responsive to different excitation wavelengths. Benefiting from this field confinement, as indicated in **Fig. R3b**, the optical field of excitation light at the spectral region of 400 - 4000 nm can be strongly enhanced by one order of magnitude or higher (for instance, 10 - 16 times in **Fig. R3b**), contributing to the ultrabroadband thermo-optical nonlinearity of GNAs.

Fig. R3 (Supplementary Figure 6.) (a) Simulations of broadband LSP in random distributed GNAs. The diameters of nanodroplets are set to be 20 - 160 nm. (b) Corresponding enhancement factor calculated by integrating the LSP field excited at different wavelengths in the random GNA in (a).

(3) The dispersion nature of the LSP of self-assembled GNAs.

Finally, we would like to emphasize that the two types of plasmonic modes (LSP and SPP) are quite different. As a localized optical mode, LSP of the Galinstan

nanodroplet can be easily excited with arbitrary incident light once the energy is matched, while the excitation of SPP requires rigorous energy and momentum conservation conditions. Therefore, unlike the SPP mode, LSP of the Galinstan nanodroplet shows no dispersion with respect to the wavevector of incident light, which means that **the dispersion nature has nothing to do with this work**, as our modulation mechanism is based on LSP resonance of nanostructure metal rather than surface plasmon polariton (SPP) of metal film.

Comment 4: “According to Fig. 1d, there are many GNAs in a nanoscale window. (1) It is for sure that the light spot size is much larger than a GNA. Therefore, multiple GNAs are simultaneously excited. Would the light beams scattering from different GNAs interfere with each other? (2) Since the GNAs are closed to each other spatially, for a typical metal plasmon wavelength in tens of nanometer level, would the signals coupling with each other? (3) Is the ‘concentration’ of the GNAs important, can the authors clarify if they can control the GNAs’ concentration, can they control the size of a GNA?”

Our Reply:

(1) **The scattering of GNA nanomaterials** may have side effect on their performance. For example, it may introduce noise to the modulation operation. However, the scattering noise of GNAs is not critical to the logical gate operation because the strong thermo-optical nonlinearity of GNAs can enable powerful logical operation overcoming the scatter noise to realize high signal-noise ratio (SNR), which have been carefully examined in the experiment.

In order to unravel quantitative influence of the scattering effect, The signal-to-noise ratio (SNR) are evaluated. In details, the central intensity contrasts of ON and OFF states are listed in **Table R1 (Table S2)** according to the modulation results of nine logical gates (**Figs. 4a,b, Supplementary Figs. 8-14**). As shown in **Table R1**, the SNR is much higher than the intensity contrast between ‘1’ and ‘0’ patterns for all the nine logical gates, allowing accurate discrimination of the logic processing.

Table R1 (Table S2) Intensity contrast and SNR of the nine logic gates

Functionality	Central intensity contrast between ‘1’ and ‘0’ patterns	SNR of output signal
AND	8:1-17:1	4.8:1
OR	2:1-10:1	21.1:1
NOT	8:1-10:1	5.1:1
NAND	10:1-16:1	31.2:1
NOR	5:1-9:1	4.1:1

XNOR	5:1-16:1	7.2:1
XOR	5:1-17:1	23.1:1
IMP	8:1-16:1	6.4:1
NIMP	8:1-17:1	6.1:1

(2) The distances among GNAs could be much smaller than tens of nanometers and some GNAs contact directly to each other. Under light illumination, a hybrid plasmonic gap mode is excited among the closed-packed GNAs to generate a macroscopic nonlinear refractive index field (RIF) through plasmon-induced resonant energy transfer (PIRET). So that the signals would couple with each other to form a diffraction ring with standard phase distribution as presented in this work.

(3) Finally, the concentration is important to the strength of plasmonic mode hybridization and thus affects the enhancement of plasmonic field. The GNAs concentration can be easily controlled by changing the mass ratio between GNA and solvent. Precise control of the size of GNAs is difficult during ultrasonication exfoliation, but we found the wide-distribution sizes of GNAs and corresponding plasmonic mode hybridization would form extremely high density of optical modes that is ideal for a broadband spectral range of LSP modulation as discussed in the reply to Comment 3.

Corresponding revisions:

In order to list the intensity contrast and SNR of the nine logic gates, we have added **Table R1** in the revised supplementary as **Table S2**.

Comment 5: “*The authors realize logical operations by controlling $\Delta\varphi$. Can the authors provide the relationship between the control beam intensity, the distance of the RIF, and the focal plane of signal beam on $\Delta\varphi$?*”

Our Reply:

Thanks for the valuable suggestions. According to the Reviewer’s suggestions, the corresponding configuration criteria of control beam strength, distance of RIF (i.e., focal position of control beam), and $\Delta\varphi$ of signal beam at focal plane for implementing nine logical gates are concluded in **Table R2 (Table 1)**. Here the strength of control beam at ‘0’ and ‘1’ states (indicated as $\Delta\varphi_{NL,0'}$ and $\Delta\varphi_{NL,1'}$) is noted as $I_{NL,0'}$ and $I_{NL,1'}$, respectively.

Table R2 (Table 1) Criteria of the nine logic gates

Gate	Focal position of control beam (with respect to the focal plane of signal beam)	$\Delta\varphi_L$	Strength contrast of control beam	$\Delta\varphi_{NL}$
AND	Behind	>0	$I_{NL'o'} > I_{NL'1'}$	$\pi > \Delta\varphi_{NL,o'} > \Delta\varphi_{NL,1'} > 0$
OR	Ahead	<0	$I_{NL'o'} < I_{NL'1'}$	$\pi > \Delta\varphi_{NL,1'} > \Delta\varphi_{NL,o'} > 0$
NOT	Ahead	<0	only $I_{NL'o'}$ (or only $I_{NL'1'}$)	$2\pi > \Delta\varphi_{NL'o'}$ (or $\Delta\varphi_{NL'1'}$) $> \pi$
NAND	Ahead	<0	$I_{NL'o'} < I_{NL'1'}$	$\Delta\varphi_{NL,1'} > \Delta\varphi_{NL,o'} > \pi$
NOR	Ahead	<0	$I_{NL'o'} < I_{NL'1'}$	$\Delta\varphi_{NL,1'} > \Delta\varphi_{NL,o'} > \pi$
XNOR	Ahead	<0	$I_{NL'o'} > I_{NL'1'}$	$\Delta\varphi_{NL,o'} > \pi > \Delta\varphi_{NL,1'} > 0$
XOR	Ahead	<0	$I_{NL'o'} < I_{NL'1'}$	$\Delta\varphi_{NL,1'} > \pi > \Delta\varphi_{NL,o'} > 0$
IMP	Ahead	<0	$I_{NL'o'} > I_{NL'1'}$	$\pi > \Delta\varphi_{NL,o'} > \Delta\varphi_{NL,1'} > 0$
NIMP	Ahead	<0	$I_{NL'o'} > I_{NL'1'}$	$\Delta\varphi_{NL,o'} > \Delta\varphi_{NL,1'} > \pi$

Comment 6: “The authors pay too much attention to the conceptual description of the all-optical logic gate in the text, and lacks a description of the specific test conditions for realizing the functions, such as the experimental environment, device parameters, the optical power and distance between the RIF and the focal point of the signal beam. It is recommended that the author supplement the experimental conditions more in details.”

Our Reply:

Thanks for the kind reminder. All the experiments were carried out at room temperature. The control and signal light sources are home-made or commercial continuous-wave solid-state lasers with wavelengths centered at 532 nm, 1342 nm, and 2.0 μm , respectively. The implementation and switching of nine logical gates are solely dependent on the strength of control beam and the distance between RIF and focal point of signal beam. The corresponding conditions are listed in **Table R3 (Table S3)**. For all the nine gates the intensity of signal beam is $\sim 0.5 - 1.5 \text{ W/cm}^2$.

Table R3 (Table S3) Main conditions of the nine logic gates

Functionality	Intensity of control beam (W/cm^2)	Distance between RIF and focal point of signal beam (mm)
AND	$<1.2 (1 + 1 = 1)$	-2.5
	$4.0 (1 + 0 = 0)$	
	$<1.2 (0 + 1 = 0)$	
	$4.0 (0 + 0 = 0)$	

OR	4.0 (1 + 1 = 1)	2.9
	2.2 (1 + 0 = 1)	
	4.0 (0 + 1 = 1)	
	2.2 (0 + 0 = 0)	
NOT	11.8 (1 → 0)	3.2
	11.8 (0 → 1)	
NAND	11.8 (1 + 1 = 0)	2.6
	7.9 (1 + 0 = 1)	
	11.8 (0 + 1 = 1)	
	7.9 (0 + 0 = 1)	
NOR	15.9 (1 + 1 = 0)	2.8
	11.8 (1 + 0 = 0)	
	15.9 (0 + 1 = 0)	
	11.8 (0 + 0 = 1)	
XNOR	4.0 (1 + 1 = 1)	4.1
	11.8 (1 + 0 = 0)	
	4.0 (0 + 1 = 0)	
	11.8 (0 + 0 = 1)	
XOR	11.8 (1 + 1 = 0)	3.4
	2.0 (1 + 0 = 1)	
	11.8 (0 + 1 = 1)	
	2.0 (0 + 0 = 0)	
IMP	4.0 (1 + 1 = 1)	3.8
	7.9 (1 + 0 = 1)	
	4.0 (0 + 1 = 0)	
	7.9 (0 + 0 = 1)	
NIMP	11.8 (1 + 1 = 0)	3.1
	15.9 (1 + 0 = 0)	
	11.8 (0 + 1 = 1)	
	15.9 (0 + 0 = 0)	

Corresponding revisions:

To enable clearer descriptions on the specific test conditions for realizing the functions, this part has been added as **Table S3** in Supplementary.

Comment 7: “Since the GNAs demonstrate strong absorption, even as high as 0.8 (SI FIG 4). That is why they must use pulsed lasers. What is the power consumption? Can the authors discuss the related limitations?”

Our Reply:

The laser sources employed for experimentally implementing all the nine AOLGs are continuous-wave solid-state lasers with wavelengths centered at 532 nm, 1342 nm, and 2.0 μm , respectively. The cross-sectional absorption of 0.8 shown in the FDTD simulation result (SI Fig. 4, which has been revised as SI Fig. 5a (Fig.R2c)) is a normalized value rather than an actual value. Here employing normalized value is beneficial to compare the plasmonic absorption behavior of Galinstan nanodroplets with different hybridization and stretchability.

The power consumption of propagation light of the 1-mm-thick GNA/NMP sample (sealed into a 1-mm-thick cuvette with an optical length of ~ 1 mm) is measured to be 13% - 15% (the total of reflection loss and absorption) in the 532 - 2000 nm region, corresponding to an insertion loss of 0.6 - 0.7 dB, much lower than the state-of-art linear-interference optical gates with phase-only spectral filter (7.5 dB) [*Nat. Commun.* 11, 5839 (2020)] and all fiber phase filters (3.5 dB) [*Nat. Commun.* 14, 1808 (2023)]. It should be noted that it is just a small step towards low powered AOLG strategy. There is still a long way and intensive efforts are needed for point-of-use low-powered large-scale optical circuit, which is thus the main limitation of such a proof-of-concept logic configuration with macroscopic absorption length. We suggest that the consumption can be significantly reduced if the AOLG scheme is further extended to nanoscale optical integration such as waveguide system with microscopic optical length of a few GNAs and highly confined control beam.

Corresponding Revisions:

The corresponding statements have been added into the revised manuscript as follows.

Line 5 in Page 16: A series of home-made and commercial continuous-wave solid-state lasers with wavelengths centered at 532 nm, 1342 nm, and 2.0 μm were employed as the light source Laser-A and Laser-B.

Line 12 in page 16: The light transmission loss of the 1-mm-thick GNA/NMP sample was measured to be 13% - 15% (the total of reflection loss and absorption) within the wavelength region of 532 - 2000 nm, corresponding to an insertion loss of 0.6 - 0.7 dB.

Comment 8: “Scale bars in all the colored maps are missing. Fonts in Fig. 4 are too small.”

Our Reply:

The scale bars have been added into the colored maps of Fig. R4 (Fig. 2b) and R5 (Fig. 4a,b,d). The fonts in Fig. R5 (Fig. 4) have been enlarged accordingly.

Fig. R4 (Fig. 2) Structure and logical states of the AOLGs based on GNAs.

Fig. R5 (Fig. 4) Experimental results of reconfigurable AOLGs based on GNA and time-dependent modulation dynamics.

Comment 9: "The title is a bit too long to read, and seems technical. I strongly suggest the authors can consider this minor point."

Our Reply: Thanks for the good advice. The title has been revised to “All-in-one, all-optical logic gates using liquid metal plasmon nonlinearity” according to the suggestion of Reviewer 3.

Responses to Reviewer #2

Overall Comments: *“Because of their fast response, all optical logic gates (AOLGs) are of great significance in large-scale integrated optical processors. This work experimentally demonstrates the ultrabroadband AOLGs enabled by reconfigurable plasmon enhanced thermo-optical nonlinearity (TONL) of liquid-metal Galinstan nanodroplet assemblies (GNAs). This is an interesting work with sufficient evidence, and I would love to see it published after minor revisions.”*

Our Reply:

We thank the Reviewer for his or her efforts on evaluating our manuscript and the quite positive comments. Based on his or her suggestions, we have carefully revised the manuscript to further improve the quality of the manuscript.

The response to the detailed comments has been listed one-by-one as follows. The corresponding changes to the original manuscript are marked in purple in the responses and in red in the main manuscript and supplementary.

Comment 1: *“The intensity at the center of a Gaussian beam is usually much stronger than the edge, so it is hard to achieve perfect destructive interference. One example is the first line of Fig. 1g. Please comment on this issue.”*

Our Reply:

We do agree that the Gaussian beam cannot support perfect destructive interference when many diffraction rings are generated at high excitation intensity (**Fig. 1g**), because the rings around the center are too dense to identify the dark center. But it is worth noting that, Fig. 1g is employed to depict the general nonlinear phenomena instead of serving as the logic component. For the proposed scheme of all-optical logical gates with 2 bits (**Fig. 2b**), **we employ extremely low excitation intensity to just generate one single diffraction ring**, as shown in Fig. R6 as well. There are at least two advantages. **On the one hand, it distinctly promotes the signal contrast. On the other hand, it essentially lowers the energy consumption** with respect to the high-power excitation condition. The control and the signal beams are the two most basic spatial modes which are easy to implement, i.e., first-order diffraction ring mode with one ring and dark center as logical state ‘0’ and fundamental Gaussian mode with bright center as logical state ‘1’ (shown in **Fig. R6**).

The logical processing is based on the reversible transition of the two modes by programmable modulating their spatial-phase shift. In this context, it only requires a small phase shift around π between the center and edge (please see the right-column

figures in **Fig. R6**). In addition, in the tightly focused region of control-signal beam modulation, interference occurs when the center and edge are focused together. Although the central region has strong light intensity, its area is very small, making the integral energy at center almost equal to that nearby the edge. Such interference is the result of overall angle integral from center to edge after focusing, and thus a quite good interference depth is achievable.

Therefore, the first-order diffraction with one ring and dark center can be easily realized as shown in the experimental section (See the AND and IMP in **Fig. R7** as examples, the other seven gates are illustrated in **Supplementary Figs. 9-15**). These signal images indicate high contrast between the central intensity of ‘1’ and ‘0’, enough for precise identification of ‘1’ and ‘0’ states (See **Table R4** for details).

Fig. R6 (Fig. 2b) Dependence of ‘1’ and ‘0’ logical states on phase shifts. $r_{\text{near-field}}$: the radial coordinates near the focal plane. $r_{\text{far-field}}$: the radial coordinates near the CCD.

Fig. R7 (Fig. 4a.b) Experimental results of reconfigurable AOLGs based on GNA and time-dependent modulation dynamics. a, Left: spatial output patterns of AND gate across visible and infrared region. Right: temporal input and output in AND gate processing. b, Left: spatial output patterns of IMP gate with broadband response. Right: temporal input and output in IMP gate processing.

Table R4 (Table S2) Intensity contrast and SNR of the nine logic gates

Functionality	Central intensity contrast between '1' and '0' patterns	SNR of output signal
AND	8:1-17:1	4.8:1
OR	2:1-10:1	21.1:1
NOT	8:1-10:1	5.1:1
NAND	10:1-16:1	31.2:1
NOR	5:1-9:1	4.1:1
XNOR	5:1-16:1	7.2:1
XOR	5:1-17:1	23.1:1
IMP	8:1-16:1	6.4:1
NIMP	8:1-17:1	6.1:1

Corresponding Revisions:

In order to list the intensity contrast and SNR of the nine logic gates, we add **Table R4** in the revised supplementary as **Table S2**.

Comment 2: “Some of the references of logic gate and refractive index in supplementary information (SI) should be included in the main text, especially those related to plasmonic method and GNAs.”

Our Reply:

Thanks for the reviewer’s kind suggestion. Some important references have been included in the main text referred as follows.

(1) References of the schemes of optical logic gate

- [1] Maram, R. et al. Frequency-domain ultrafast passive logic: NOT and XNOR gates. *Nat. Commun.* **11**, 5839 (2020).
- [4] Li, Y. et al. Nonlinear co-generation of graphene plasmons for optoelectronic logic operations. *Nat. Commun.* **13**, 3138 (2022).
- [7] Piccione, B., Cho, C. H., van Vugt, L. K. & Agarwal, R. All-optical active switching in individual semiconductor nanowires. *Nat. Nanotechnol.* **7**, 640-645 (2012).
- [8] Yang, H. et al. Nanowire network-based multifunctional all-optical logic gates. *Sci. Adv.* **4**, eaar7954 (2018).
- [9] Hendra, et al. Photochemically switchable interconnected microcavities for all-organic optical logic gate. *Adv. Funct. Mater.* **31**, 2103685 (2021).
- [10] Kaushal, S. et al. All-fibre phase filters with 1-GHz resolution for high-speed passive optical logic processing. *Nat. Commun.* **14**, 1808 (2023).
- [11] Wu, L. et al. All-optical logic devices based on black arsenic-phosphorus with strong nonlinear

- optical response and high stability. *Opto-Electron. Adv.* **5**, 200046 (2022).
- [12] Sk, K. et al. Nonlinear coherent light-matter interaction in 2D MoSe₂ nanoflakes for all-optical switching and logic applications. *Adv. Opt. Mater.* **10**, 2200791 (2022).
- [13] Song, C. et al. Liquid phase exfoliated boron nanosheets for all-optical modulation and logic gates. *Sci. Bull.* **65**, 1030-1038 (2020).
- [49] Boolakee, T. et al. Light-field control of real and virtual charge carriers. *Nature* **605**, 251-255 (2022).
- [50] Zasedatelev, A. V. et al. A room-temperature organic polariton transistor. *Nat. Photon.* **13**, 378-383 (2019).
- [51] Kim, J. et al. Photon-triggered nanowire transistors. *Nat. Nanotechnol.* **12**, 963-968 (2017).
- [52] Wei, H., Wang, Z., Tian, X., Käll, M. & Xu, H. Cascaded logic gates in nanophotonic plasmon networks. *Nat. Commun.* **2**, 387 (2011).
- [53] Sang, Y. et al. Broadband multifunctional plasmonic logic gates. *Adv. Opt. Mater.* **6**, 1701368 (2018).

(2) References on the nonlinear refractive index of nanomaterials

- [36] Zhang, J. et al. Broadband spatial self-phase modulation of black phosphorous. *Opt. Lett.* **41**, 1704-1707 (2016).
- [37] Wang, G. et al. Tunable effective nonlinear refractive index of graphene dispersions during the distortion of spatial self-phase modulation. *Appl. Phys. Lett.* **104**, 141909 (2014).
- [38] Lu, L. et al. Broadband nonlinear optical response in few-layer antimonene and antimonene quantum dots: A promising optical Kerr media with enhanced stability. *Adv. Opt. Mater.* **5**, 1700301 (2017).
- [39] Wu, L. et al. Perovskite CsPbX₃: A promising nonlinear optical material and its applications for ambient all-optical switching with enhanced stability. *Adv. Opt. Mater.* **6**, 1800400 (2018).
- [40] Wu, L. et al. Few-layer tin sulfide: A promising black-phosphorus-analogue 2D material with exceptionally large nonlinear optical response, high stability, and applications in all-optical switching and wavelength conversion. *Adv. Opt. Mater.* **6**, 1700985 (2018).
- [41] Lu, L. et al. Few-layer bismuthene: Sonochemical exfoliation, nonlinear optics and applications for ultrafast photonics with enhanced stability. *Laser Photon. Rev.* **12**, 1700221 (2017).
- [42] Shan, Y. et al. Spatial self-phase modulation and all-optical switching of graphene oxide dispersions. *J. Alloys Compd.* **771**, 900-904 (2019).
- [43] Wang, G. et al. Tunable nonlinear refractive index of two-dimensional MoS₂, WS₂, and MoSe₂ nanosheet dispersions. *Photonics Res.* **3**, A51-A55 (2015).
- [44] Shan, Y. et al. A promising nonlinear optical material and its applications for all-optical switching and information converters based on the spatial self-phase modulation (SSPM) effect of TaSe₂ nanosheets. *J. Mater. Chem. C* **7**, 3811-3816 (2019).
- [45] Shi, B. et al. Broadband ultrafast spatial self-phase modulation for topological insulator Bi₂Te₃ dispersions. *Appl. Phys. Lett.* **107**, 151101 (2015).
- [46] Li, X. et al. Tri-phase all-optical switching and broadband nonlinear optical response in Bi₂Se₃ nanosheets. *Opt. Express* **25**, 18346-18354 (2017).

Comment 3: “Is it the same to use Au Nps with different size? Also, the solid-state Au Nps might be better for integration than the liquid-state GNAs. By the way the discussion in SI about possible integration of GNAs is not sufficient.”

Our Reply:

(1) Unique advantages of liquid-state GNAs

This scheme of all-in-one AOLGs is also suitable for Au NPs and other nanostructured plasmonic metals. However, here the implementation of ultrabroadband (532-2000 nm in experiment, 400-4000 nm in potential) and all-in-one AOLGs benefits from the ultrabroadband and strong localized-surface-plasmon resonance of GNAs, which is stemmed from their unique merits, such as (i) giant intrinsic absorption due to much large imaginary part of dielectric function, (ii) easy fabrication for heterogenous plasmonic nanostructures with broad distributed sizes, (iii) strong enhanced nonlinearity at ultrabroad spectral range induced by the excellent inherent stretchability of liquid metal, (iv) great potential for realizing flexible all-optical logic element. These merits can be discussed in detail as follows.

(i) GNAs possesses giant intrinsic absorption property.

The intrinsic light absorption nature of GNAs is highly dependent on the dielectric function of the material. **Fig. R8** clearly shows the direct comparison of the measured dielectric functions (ϵ_r : real part, ϵ_i : imaginary part) between the as-synthesized bulk Galinstan droplets and noble metal Au and Ag. Galinstan exhibits comparable ϵ_r to Au and Ag, while ϵ_i of Galinstan is one or two orders larger, implying that Galinstan is a superior plasmonic material for broadband light harvesting with much higher figure of merit ($-\epsilon_i/\epsilon_r$) with respect to conventional plasmonic metals [*Nature* 581, 401-405 (2020)] (see **Fig. R9** for details). These intriguing characteristics indicate versatile potential of Galinstan in opto-thermal based optical applications.

Fig. R8 (Fig. 1b,c) (b) Real and (c) imaginary part of the dielectric constant of Galinstan, Ag and Au.

Fig. R9 (Supplementary Figure 1) Figure of merit ($-\epsilon_i / \epsilon_r$) for plasmonic absorption of bulk Galinstan liquid.

(ii) GNAs is favorable for broadband light absorption due to the unique heterogeneous nanostructures.

Due to the much weaker bond energy in liquid metal than that in solid metal, GNAs with broad-distributed sizes in nanoscale can be prepared by simple ultrasonication exfoliation of bulk droplets in solvent (**Fig. R10**). As illustrated in **Fig. R11**, GNAs with sizes ranging from ~ 30 to 150 nm and versatile assembly configuration can be conveniently obtained, beneficial for high density of plasmonic modes and resonant interparticle coupling. Specially, it is also important to point out a unique feature of GNAs different from Au and other solid metal, i.e., the self-limiting oxidation of gallium. As shown in the inset of **Fig. 1e**, the prepared GNAs are conformably wrapped with insulation nanolayers (~ 2 nm thickness) of Ga_2O_3 [*ACS Appl. Mater. Interfaces* 6, 18369 (2014); *Nano Lett.* 11, 5104 (2011)], forming a self-assembling Galinstan/ Ga_2O_3 core-shell framework. The self-limiting oxide layers can effectively prevent electron transfer among adjacent nanodroplets, providing a robust shield for keeping strong surface electron localization and stable plasmonic effects.

Fig. R9 (Supplementary Figure 2) Synthesis of GNAs.

Fig. R10 (Fig. 1e,d) d, TEM of GNAs. e, Corresponding size distribution counting of GNAs (Inset: Magnified TEM for single Galinstan nanodroplet. The dashed line notes the boundary of Ga₂O₃ shell).

(iii) Liquid GNAs can be more advantageous for plasmon hybridization and optical enhancement due to unique mechanical flexibility.

The excellent inherent stretchability of liquid phase makes the GNAs tend to be morphed into massive irregular geometries under stress and gravity [*Mater. Sci. Eng. R-Rep.* 138, 1-35 (2019)] as presented in **Fig. R11**. It leads to strong resonant interparticle hybridizations with local hot-spot field enhancements and additional new resonant mode formation [*Science* 302, 419-422 (2003)], which further greatly enhance the LSP nonlinearity and broaden the absorption band.

Fig. R11 (Supplementary Figure 3e) GNAs morphed into massive irregular geometries.

(iv) Liquid GNAs can be ideal candidates for flexible optical devices.

Moreover, relying on the unique advantage of flexible shape morphing of GNA, flexible all-optical logic element, which is the key component for future flexible photonic circuits, is also predictable by integrating GNAs with soft photonic materials.

(2) Possibility of nanoscale modulation via a few GNAs

We agree that the discussion on integration in the original Supplementary Content VI is not sufficient for actual waveguide system, which have been deleted in the revised edition. As an alternative demonstration, we theoretically investigate whether the proposed TONL modulation scheme can work on nanoscale, because it is the primary question should be solved to verify the possibility of integrated AOLG. The simulation is based on a few GNAs of 400-nm thickness and 532-nm wavelength of signal and control beams. Other simulation parameters are listed in **Table R5**.

The simulation on signal-beam evolution in nanoscale TONL modulation (shown in **Fig. R12**) clearly demonstrates that the nonlinear refractive index field within nanoscale distribution, which is induced by highly confined control beam, can enable efficient modulation on the spatial mode of signal beam. This modulation strongly depends on the change of refractive index, local range of nonlinear refractive index field, and distance of focus points between control and signal beams. These results can serve as the evidence as the feasibility of the proposed AOLG scheme on miniaturized integration. These results suggest the feasibility of the proposed AOLG scheme on miniaturized integration.

Fig. R12 (Supplementary Figure 16) Simulation of signal-beam evolution during the nanoscale all-optical SXPM modulation based on a few GNAs with 400-nm thickness. (a) At different modulation depths of the refractive index. (b) At different local ranges of refractive index field. (c) At different positions of the refractive index field. d) Schematic of waveguide-based AOLG device. Black dotted circle: local range of refractive index field. Δn : change of

refractive index at the waist of control beam; w_c : waist radius of control beam; d_{c-s} : distance of focus points between control and signal beams.

Table R5 (Table S5) Simulation parameters of Supplementary Figure 16

Figure	Control beam	Signal beam	System
a(I)	$w_c = 500$ nm $I_c = 0$ W/cm ² $d_{c-s} = 2.2$ μm		
a(II)	500 nm 12 kW/cm ² 2.2 μm		
a(III)	500 nm 24 kW/cm ² 2.2 μm		
b(I)	200 nm 12 kW/cm ² 2.2 μm		Wavelength: 532 nm
b(II)	500 nm 12 kW/cm ² 2.2 μm	$w_s = 500$ nm $I_s < 10$ kW/cm ²	Thickness of few GNAs: 400 nm
b(III)	1000 nm 12 kW/cm ² 2.2 μm		$n_0 = 1.47$ $n_2 = 1.68 * 10^{-5}$ cm ² /W
c(I)	500 nm 12 kW/cm ² -1.8 μm		
c(II)	500 nm 12 kW/cm ² 0.2 μm		
c(III)	500 nm 12 kW/cm ² 2.2 μm		

w_c : waist radius of control beam; I_c : intensity of control beam; d_{c-s} : distance of focus points between control and signal beams; w_s : waist radius of signal beam; I_s : intensity of signal beam

Corresponding Revisions:

Fig. R12 and **Table R5** with corresponding descriptions have been added into the revised supplementary Content VI as **Supplementary Figure 16** and **Table S5**, respectively. The title of Supplementary Content VI has been revised from “Nano-AOLGs via GNAs” to “**Possibility of nanoscale modulation via a few GNAs.**”

Comment 4: “In SI it was said: Constructive interference occurs when the phase difference equals to $(2m-1)$ ($m \in \mathbb{Z}$), with the appearance of bright ring referred to as m th-order diffraction ring mode. Is it correct?”

Our Reply:

Many thanks for the valuable correction. This sentence has been corrected as follows.

In Page 8 of supplementary: Constructive interference occurs when the phase difference $|\Delta\phi(0) - \Delta\phi(\infty)| \approx (2N-1)\pi$ ($N = 1, 2, 3, \dots$) with the appearance of bright ring referred to as N th-order diffraction ring mode (N is also the ring number). If $|\Delta\phi(0) - \Delta\phi(\infty)|$ is much larger than π , then a series of concentric interference rings appear.

Comment 5: “In SI Figure 8 and subsequent figures, the sequence in (a) and (b) is not consistent: $0+0, 0+1, \text{etc.}$ ”

Our Reply:

We greatly thank the Reviewer for the careful evaluation and point out the typos. In the revised supplementary information, we have carefully proofed and corrected **Supplementary Figures 9-15**. Accordingly, the revised **Fig. R13 (Supplementary Figure 9)** is shown below as an example. Please see revised supplementary for detailed **Supplementary Figures 9-15**.

Fig. R13 (Supplementary Figure 9) Design and experimental results of OR gate. (a) Mechanism of phase modulation and corresponding scheme of phase dependence. (b) Left: spatial output patterns of OR gate across visible and infrared regions. Right: temporal input and output in processing.

Comment 6: “For all the logic gates, please give the contrast between state 1 and state 0. Also, please give the SNR.”

Our Reply: The contrasts between state 1 and 0 as well as the SNR have been measured according to the Reviewer’s suggestion, which are listed in **Table R6**.

Table R6 (Table S2) Intensity contrast and SNR of the nine logic gates

Functionality	Central intensity contrast between ‘1’ and ‘0’ patterns	SNR of output signal
AND	8:1-17:1	4.8:1
OR	2:1-10:1	21.1:1
NOT	8:1-10:1	5.1:1
NAND	10:1-16:1	31.2:1
NOR	5:1-9:1	4.1:1
XNOR	5:1-16:1	7.2:1
XOR	5:1-17:1	23.1:1
IMP	8:1-16:1	6.4:1
NIMP	8:1-17:1	6.1:1

Corresponding Revisions:

Table R6 has been added to revised supplementary as **Table S2**. The corresponding description on **Table S2** has been added into the revised manuscript as follows.

Line 9 in Page 12: These AND images indicate a high contrast of the central intensity between the states of ‘1’ and ‘0’ in the range of 8:1-17:1, enough for precise identification of ‘1’ and ‘0’ states (see **Table S2** for detail).

Line 19 in Page 12: Accurate discrimination of the logic processing is feasible since the signal-to-noise ratios (SNRs) of the nine logic gates are higher than 4.1:1 (see **Table S2** for detail), comparable with the performance of optoelectronic logic gates based on graphene plasmons⁴.

[4] Li, Y. et al. Nonlinear co-generation of graphene plasmons for optoelectronic logic operations. *Nat. Commun.* **13**, 3138 (2022).

Responses to Reviewer #3

Overall Comment: *“The authors demonstrate a reprogrammable/multifunctional optical logic gate (from 532-2000 nm) based on spatial cross phase modulation using liquid-metal Galinstan nanodroplets....The experiments are thorough and well done, including careful characterization of the nanodroplets and their complex nonlinear susceptibility. As far as I can tell, the science is solid and sound. Figures are well designed and beautiful (really nice!). I also appreciate the long supplement that gives detailed background on the work.”... “Though the work is clever and executed well, ... I don't think this work would be attractive to a broad audience or have a broad impact.... I also comment on revisions that should be addressed if the editors and other reviewers wish to see the work published in Nature Communications...”*

Our reply:

We thank the Reviewer for his or her great efforts on evaluating our manuscript. We are delighted to see that the science is regarded as “solid and sound”, the figures are “really nice”, and the overall work is “clever and executed well”, indicating the novelty and advancement of the proposed liquid metal based AOLG for the field of optical signal processing. We also sincerely appreciate the comment on the broad impact in the view point of the information and communication technology (not just limited in the optical community).

In order to promote the significance or impact of the work, carefully point-to-point replies and related revisions have been made according to the detailed comments as follows, which do have essentially improved the quality of the manuscript. The corresponding changes to the original manuscript are marked in purple in the responses and in red in the main manuscript and supplementary.

Comment 1: *“The field of optical signal processing has rightly come under intense scrutiny over the last decade due to practical bottlenecks in information and communication technology, for which simply switching them to optical systems won't address.” “And certainly, simply switching from electronic systems to optical is not the way to speed up information- which unfortunately is an argument that many optical scientists still make.” “The use of optical processing is of great importance still, but it's more nuanced: as it relates to networking and transmission, interconnects, and most definitely quantum information (likely the future of computation and communication).”*

“We know that the major cost and problems with the speed of information is energy consumption and heat dissipation. Current commercial CMOS electronics is much more energy efficient than any optical system shown to date. Electronic transistors consume

~0.1 fJ per bit. At speeds of ~10 GHz, this amounts to a microwatt of average power. Optics, especially when nonlinearity is employed is energy greedy. The best nonlinear optical gates to date are at the ~1 pJ per bit level, 10,000x worse for energy than current commercial CMOS. Optical gates using linear interference have shown much less energy consumption, but have other issues- integration, fan-out, scalability, miniaturization. The moral of the story is that optical signal processing for optical processing sake is really not broadly interesting anymore.”

“The authors first statement in the abstract is, ‘Multifunctional all-optical logic gates (AOLGs) of massively parallel processing are of great importance for large-scale integrated optical processors with executing speed far in excess of electronics, while are rather challenging due to limited operation bandwidth and multifunctional integration complexity.’ That’s fair. They are being honest. However, this implies that large-scale integrated optical processors are interesting in their own right. This work therefore may or may not (it’s research after all) help people mainly in the field of optical processing. If this work could be applied more broadly to information processing, solve an issue with energy consumption for high-speed signals, applied to quantum information processing, or nanoscale optical integration, this would have a broader impact and warrant publication in a scientific journal with readership of broad scientific backgrounds.”

Our Reply:

We sincerely thank the Reviewer for his or her idea exchange on the roadmap of information technology, which is the ultimate pursuit of both electronic and optical communities in the post Moore era. The tradeoff among modulation speed, energy consumption and heat sink are currently hindering the development of electronic systems, which would definitely be going to playing crucial roles in the practical pathways for optical systems. However, it doesn’t mean that the endeavors from optical community are out of the roadmap or make no sense at its current status. Taking the nonlinear optical strategies for example, the proposed AOLG is trying to provide the **proof-of-concept scheme for all-in-one logic functionality with unique advantages of ultrabroad operation bandwidth as well as extra bonus on reduced energy consumption**. A more detailed discussions are attached as follows.

(1) Unique advantages of optical circuits beyond electronics

The physical limitations of electronic transistors in processing speed, bandwidth capabilities, and their energy consumption will do have push the Moore’s Law to come to an end in the future, which in turn inspire and accelerate a variety of newly emerging technologies including optical information processing [*Nat. Photon.* 4, 345 (2010); *Nature* 605, 251-255 (2022)], atomic Lego-like electronics [*Nature Electronics* 5, 327-

328 (2022)], liquid-phase DNA computing [*Nature* 622, 292-300 (2023)], microfluidic transistor [*Nature* 622, 735-741 (2023)], etc. However, it should be recognized that the **development of optical circuits has been starting later and/or slower than that of its electronic counterparts**. One of the most crucial challenges is the lack of high-performance optical logic components which can bring unique advantages that electronics cannot do, as highly perused in other alternative post-Moore-era technologies in their primary stages. The foundation of optical processing is to utilize all the inherent characteristics of light, particularly light speed and remarkable broad spectral/frequency bandwidth, to process high density of information at high speed and low energy consumption. Therefore, massively parallel processing based on broad operation bandwidth is widely regarded as one of the main advantages of optical computing compared to its electronic counterpart which mainly uses serial processing. In addition, realizing both high speed and low energy consumption beyond the limitations of electronic circuit is also the critical challenge to the implementation and development of optical computing. Solving these practical bottlenecks requires innovation of fundamental optical technologies, but **such innovations should only pushed forward step by step that needs incremental development of optical logics** under contribution of all the researchers.

(2) Growing attraction of optical processing

Optical processing does still attract broad interests nowadays, during which a variety of optical strategies have been pushing on their way towards the ultimate goal of modern information technologies. In recent years, many groups have reported the latest development of optical logic gates based on linear and nonlinear optic effects, including **optoelectronic effects of graphene-based driven by femtosecond laser pulses** [*Nature* 605, 251-255 (2022); *Nat. Commun.* 13, 3138 (2022)], **optoelectronic effects of perovskite photodetector** [*Nat. Commun.* 13, 720 (2022)], **nonlinear optical effects or photoluminescence of semiconductor nanowires** [*Nat. Commun.* 2, 387 (2011); *Nat. Nanotechnol.* 7, 640-645 (2012); *Nat. Nanotechnol.* 12, 963-968 (2017); *Sci. Adv.* 4, eaar7954 (2018)], **nonlinear optical effect in semiconductor microcavity** [*Nat. Photon.* 13, 378-383 (2019)], **nonlinear optical effect with chirality degree of freedom in MoS₂** [*Sci. Adv.* 8, eabq8246 (2022)], and **linear interference with phase linear filtering** [*Nat. Commun.* 11, 5839 (2020); *Nat. Commun.* 14, 1808 (2023)]. These latest works demonstrate vast advancements of optical logics and indicate the great potential of optical information processing. Under these efforts of optical scientists, optical computing continues to develop with achievements benefitting a lot of frontier scientific topics such as nano-photonics, bio-photonics, meta-optics, optical neural networks, and quantum information processing and communication.

(3) Current motivation for creating broadband all-in-one optical logics

We also agree with the reviewer that these reported strategies confront a lot of challenging issues, including high-latency and cumbersome optic-electric conversion of common photoelectric effect, high energy consumption and cost of femtosecond laser, and large footprint of linear interference components, as well as power hungry of nonlinear effects, etc. In particular, **most strategies have yet to show broad operation bandwidth (only lay within tens of nanometers) or superior compatibility** of multifunctional integration, shadowing their potentials in massively parallel processing.

Since large operation bandwidth is one of the most unique advantages of optical logics that electronic logics don't have, how to implement it has to be addressed in order to make the dream of high-speed processing of high-density optical data come true. Even more, the practicality of large-scale integrated optical processor naturally implies the demand of multifunctional all-optical logic gate (AOLG) components. Although combination of AND, OR, and NOT gates (or combination of NAND and NOR) is a universal approach to perform all other logical gates in electric circuits, it is impractical for large-scale optical circuits because the diffraction limit of light is a fundamental obstacle for reducing the dimensions of optical logic components to the length scales of electronic ones. When it comes to large-scale optical circuits, combining and cascading too many AOLG units will surely confront the shortcomings of high latency, high energy consumption, complex structure, low scalability, and difficult fabrication and miniaturization. For these reasons, **it is important to realize a nanoscale AOLG with both broad operation bandwidth and multiple logical functions as a promising building block of future all-optical nano-processor.**

(4) Significances and prospects of our work

Finally, we re-emphasize that, in this work, by exploiting the thermo-optical nonlinearity (TONL) of liquid-metal flexible nanodroplets, we **for the first time experimentally demonstrate a reconfigurable all-in-one AOLG that achieves nine fundamental Boolean logics in a single configuration with ultra-broad operation bandwidth (532 - 2000 nm in experiment, 400 - 4000 nm in potential).** The TONL is strongly enhanced by the ultrabroadband plasmonic field confinement of nanodroplets that endows this AOLG a series of extraordinary advantages, including high contrast between '1' and '0' states, low signal-to-noise ratios, and fast processing speed (~210 ps).

In particular, the light transmission loss of the 1-mm-thick nanodroplet dispersion is only 13%-15% (the total of reflection loss and absorption) in the 532 - 2000 nm region, corresponding to an insertion loss of 0.6 - 0.7 dB, lower than the state-of-art linear-interference optical gates with phase-only spectral filter (7.5 dB) [Nat. Commun.

11, 5839 (2020)] and all fiber phase filters (3.5 dB) [Nat. Commun. 14, 1808 (2023)].

Furthermore, with extra numerical simulation, we delineate that **nanoscale TONL effect of a few nanodroplets is capable of enabling the proposed broadband AOLG scheme in nanoscale optical integration**. In this context, higher performance can be expected because nanoscale integration will induce stronger plasmonic field confinement. Therefore, our work may provide a potential avenue for solving some practical bottlenecks that limit the long-anticipated revolutionary capabilities of optical circuits. Furthermore, it is reasonable to argue that the development of optical processing will never stall, and the great potential of broadband all-optical circuits in massively parallel processing will have a central role in driving future information technologies.

Suggestions for Revision:

Comment 2: *“If the editor disagrees with my assessment of lack of broad impact and decides to publish this work in Nat. Comm, the motivation (abstract and introduction) needs to start much broader. It should be laid out as*

-Why high-speed logic is interesting to information processing and communications broadly (not just optics or plasmonics)

-What are bottlenecks in high-speed logic (not just plasmonics)

-What have others done to solve this problem (not just nonlinear optics)

-What are the shortfalls they are not meeting: lack of single platform multifunctional device, small wavelength operating range, small scale integration, etc.

-Then how the technique using plasmon enhanced liquid metal nanodroplets solves it”

Our Reply:

Thanks so much for these critical suggestions. According to the suggested outline, the abstract and introduction have been revised as follows.

Revised Abstract: *Electronic processors are reaching the physical speed ceiling that heralds the era of optical processors.* Multifunctional all-optical logic gates (AOLGs) of massively parallel processing are of great importance for large-scale integrated optical processors with speed far in excess of electronics, while are rather challenging due to limited operation bandwidth and multifunctional integration complexity. Here we for the first time experimentally demonstrate a reconfigurable all-in-one broadband AOLG that achieves nine fundamental Boolean logics in a single configuration, enabled by ultrabroadband (400 - 4000 nm) plasmon-enhanced thermo-optical nonlinearity (TONL) of liquid-metal Galinstan nanodroplet assemblies (GNAs). Due to the unique heterogeneity (broad-range geometry sizes, morphology, assembly profiles), the prepared GNAs exhibit broadband plasmonic opto-thermal effects (hybridization, local

heating, energy transfer, etc.), resulting in a huge nonlinear refractive index under the order of 10^{-4} - 10^{-5} within visual-infrared range. Furthermore, a generalized control-signal light route is proposed for the dynamic TONL modulation of reversible spatial-phase shift, based on which nine logic functions are reconfigurable in one single AOLG configuration. Our work will provide a powerful strategy on large-bandwidth all-optical circuits for high-density data processing in the future.

Revised Introduction: Computer technology based on high-speed logics is the cornerstone of modern information processing and communications. Nevertheless, current computer processors built by electronic circuits may confront the physical limitation to continue Moore's law within next two decades¹. Future development of computer technology requires new principles and technologies. Photonic circuit is now widely regarded as one of the most potential successors to its electronic counterparts because optical signal processing has a series of advantages such as ultrahigh bit-rate, large bandwidth, great concurrency, as well as ultralow cross-talk²⁻⁴. Nowadays, the practical implementation of optical processing is based on optic-electric interconnection in which the digital signal is processed in electronic processor and the light is applied as signal transmitter. However, its performance is far from the speed ceiling of optical computation since the restriction of high-latency and cumbersome optic-electric conversion. In this respect, it is of great importance to progress all-optical system by replacing the entire electronic functional components with all-optical elements^{5,6}. The promising building blocks for optical processors are the AOLGs, which can enable logic functions by manipulating the intensity, phase, polarization, or wavelength of optical signals through light-matter interaction.

Among the intensive research of AOLGs in the past decades, bandwidth scalability and multifunctionality are among the most severe challenges determining the feasibility of high-speed optical processors^{3,6}. The practicality of optical processors implies the urgent demand of multifunctional AOLG components since different optical logic gates will need to be made physically different. This will give different spatial modes or entail many components having different spatial mode requirements, now making each component bulky, unique, and not easily printable in a universal way like electronics. The past decade has witnessed vast advancements in AOLGs based on linear or nonlinear optical modulation, ranging from stimulated scattering and photoluminescence (PL) of nanowires or nanospheres⁷⁻⁹, linear interference with phase-only linear filtering or plasmonic gratings^{1,10}, to spatial self-phase modulation (SSPM) or spatial cross-phase modulation (SXPM) of nanosheets¹¹⁻¹³. However, most of these strategies reported thus far have yet to show wide operation bandwidth (only lay within tens of nanometers) or superior compatibility of multifunctional integration, shadowing

their potential for massively parallel processing. In this case, more rational strategies have been long pursued for the further development towards high scalability and multifunctionality.

The intriguing optical nonlinearity arisen from the surface plasmonics of nanostructured metallic materials provides a new approach to light manipulation in nano-optical devices¹⁴⁻¹⁸. As a special class of metal, room-temperature liquid metal alloys typified by Galinstan and eutectic GaIn have been of great interest nowadays due to their stable liquid phase, exceptional stretchability, strong plasmonic effect, high thermal conductivity, and electrical conductivity, together with biocompatible low toxicity compared with mercury¹⁹⁻²¹. These unique properties imply great potential of liquid-metal-based plasmonic nanostructures for exploring advanced AOLGs, yet the corresponding strategy remains undiscovered.

In this work, we exploit the plasmon-enhanced TONL of liquid-metal GNAs to demonstrate a reconfigurable all-in-one AOLG based on dynamic and programmable manipulation on the reversible phase shift of dual-beam SXPM interaction. In such a single all-optical configuration without external electronic modulation, nine fundamental Boolean logics are achievable including AND, OR, NOT, NOR, NAND, XNOR, XOR, material implication (IMP), and not material implication (NIMP). The ultrabroadband (400 - 4000 nm) plasmon-enhanced light harvesting of GNAs endows a logic operation band ranging from visible to infrared regions. Our results would provide a promising strategy to overcome the limitation of bandwidth and multifunctionality in traditional AOLG schemes, inspiring a new pathway towards optical processor.

Other Revisions:

The two prestigious references about linear-interference optical logic gates have been added in the revised reference list.

[1] Maram, R. et al. Frequency-domain ultrafast passive logic: NOT and XNOR gates. *Nat. Commun.* **11**, 5839 (2020).

[10] Kaushal, S. et al. All-fibre phase filters with 1-GHz resolution for high-speed passive optical logic processing. *Nat. Commun.* **14**, 1808 (2023).

Comment 3: *In the beginning of the discussion section the authors claim that the technique could be scaled to the micro-optic, on-chip level. The authors give some proposed details: coating waveguides with GNAs, ion beam injection, using MEMs mirrors to steer beams, etc. They give some simulation results of their technique at small scale, however, not in a prototype waveguide system.*

The simulation in the supplement therefore is too simple/superficial at this point to be included and there is a lot of guessing what this system might look like. The authors

should remove their simulation results until they have a simulation in an actual waveguide structure like what they show in S15 d) as well as dial down scaling claims. The simulation is a good start, just not compelling enough now.

The cuvette the authors use in their proof of concept is 1 mm thick. That's pretty macroscopic. Would just a few GNAs actually work in a scaled down version as they say? What is the optical thickness of GNAs used in the simulation? This is not mentioned and an important detail.

Note: it may be fair to make some gentle/softer claims about scaling down to chip level- 1-3 sentences with references. Or it could be o.k to include the simulation IF much, much more details of the simulation parameters are given. Regardless, the schematic of the proposed circuit, S15d should be removed. This schematic is a nice idea but not tested at all by the authors' simulation or backed up by reference.

Our Reply:

(1) Modified discussion on the scaling and integration of the AOLG scheme.

We greatly appreciate the valuable comments, especially the constructive suggestions on few-GNAs simulation models, which are favorable for the feasibility discussions on the nanoscale level integration.

Based on the reviewer's professional comments, we understand that the simulation of the nanoscale modulation via a few GNAs in free space (presented in **Supplementary Figs. 15a-c**) is still not sufficient for proving the assumed integrated waveguide scheme (**Supplementary Fig. 15d**). Since then, we have tried our best to simulate the AOLG operation in an actual waveguide structure, but after five months of trying we recognize that it should be a complex and long-term work in the future, because the phase modulation for near-field interference under tight focusing in waveguide has high complexity and we could not find any related literatures to refer to.

Therefore, **In the revised version, we have deleted Supplementary Fig. 15d.** Instead, according to the Reviewer's suggestion, **we turn to focus on theoretically investigating whether the proposed TONL modulation scheme with a few GNAs can work on nanoscale**, considering that it is the primary question should be solved to study the possibility of integrated all-in-on AOLG.

The simulation on signal-beam evolution in nanoscale TONL modulation with 400-nm-thick GNAs (shown in **Fig. R14**) clearly demonstrates that the nonlinear refractive index field within nanoscale distribution, which is induced by plasmonic opto-thermal effects of a few GNAs, can enable efficient modulation on the spatial mode of signal beam just as in the macroscopic system experimentally shown in this work. This microscopic modulation strongly depends on the change of refractive index, local range of nonlinear refractive index field, and distance of focus points between control and signal beams. **These results suggest the feasibility of the proposed AOLG scheme**

on miniaturized integration, considering the corresponding guided-wave propagation for both the fundamental Gaussian beam mode and first-order diffraction ring mode can be supported in a ring-core waveguide structure [*Phys. Rev. Lett.* 121, 233602 (2018)]. We will also try our best to develop our technology to actually fabricate the waveguide-based device and verify its feasibility in further research.

Fig. R14 (Supplementary Figure 16) Simulation of signal-beam evolution during the nanoscale all-optical SXPM modulation based on a few GNAs with 400-nm thickness. (a) At different modulation depths of the refractive index. (b) At different local ranges of refractive index field. (c) At different positions of the refractive index field. (d) Schematic of waveguide-based AOLG device. Black dotted circle: local range of refractive index field. Δn : change of refractive index at the waist of control beam; w_c : waist radius of control beam; d_{c-s} : distance of focus points between control and signal beams.

(2) Parameters of the numerical simulation

The numerical simulation of **Fig. R14** is based on a few GNAs of 400-nm thickness and 532-nm wavelength of signal and control beams. Other simulation parameters are listed in **Table R7**. Please note that the intensities of signal and control beams in the simulation are around the order of 10 kW/cm², much higher than that of the

experimental parameter with 1-mm cuvette. That is because the focused beam in the simulation has a much short Rayleigh length that leads to a high intensity threshold (~ 10 kW/cm²) for ‘0’-‘1’ signal mode transition. But such an intensity level is conventional for guided light in nanophotonic waveguide. Larger waist radius with longer Rayleigh length can lower the intensity threshold.

Table R7 (Table S5) Simulation parameters of Supplementary Figure 16

Figure	Control beam	Signal beam	System
a(I)	$w_c = 500$ nm $I_c = 0$ W/cm ² $d_{c-s} = 2.2$ μ m		
a(II)	500 nm 12 kW/cm ² 2.2 μ m		
a(III)	500 nm 24 kW/cm ² 2.2 μ m		
b(I)	200 nm 12 kW/cm ² 2.2 μ m		Wavelength: 532 nm
b(II)	500 nm 12 kW/cm ² 2.2 μ m	$w_s = 500$ nm $I_s < 10$ kW/cm ²	Thickness of few GNAs: 400 nm
b(III)	1000 nm 12 kW/cm ² 2.2 μ m		$n_0 = 1.47$ $n_2 = 1.68 \cdot 10^{-5}$ cm ² /W
c(I)	500 nm 12 kW/cm ² -1.8 μ m		
c(II)	500 nm 12 kW/cm ² 0.2 μ m		
c(III)	500 nm 12 kW/cm ² 2.2 μ m		

w_c : waist radius of control beam; I_c : intensity of control beam; d_{c-s} : distance of focus points between control and signal beams; w_s : waist radius of signal beam; I_s : intensity of signal beam

Corresponding Revisions:

Fig. R18 and **Table R7** with corresponding descriptions have been included in the

revised supplementary as **Supplementary Figure 16** and **Table S5**, respectively.

(3) Revised statements on potential applications for integration

The statements about the potential integration and chip-level applications have been revised to be more gentle and softer as follows.

Corresponding Revisions:

The statement “Here we demonstrate the AOLG scheme compress to the nanoscale **based on single or a few GNAs inserted waveguide systems** in numerical simulation (Supplementary Section VI), exploring the **feasibility of the waveguide or flexible integration platform**. Apart from the proof-of-concept demonstration in the solution-based devices, we suggest that the proposed broadband AOLG route can be flexibly extended to **the on-chip platforms, which are crucial for next generation high density optical integration.**” has been revised as:

Line 14 in page 14: “Based on numerical simulation, here we further delineate that nanoscale TONL effects of **a few GNAs** are capable of modulating the beam spatial mode through SXPM just as in the macroscopic system experimentally shown in this work (see Supplementary Section VI for details), suggesting the feasibility of the broadband proposed AOLG scheme **in microscopic system**. Therefore, apart from the proof-of-concept demonstration in the dispersion-based devices, this scheme may be applicable to **nanoscale optical integration such as waveguide system.**”

Comment 4: *Order is a little mixed in my opinion in the first few sections. I would suggest explaining the concept of the device generally first (beam with bright center/beam with dark center) then give background on SXPM and the nanodroplets. In fact, the section characterizing the nanodroplets really should be moved to the supplement. As it is, it is confusing to have a lot of details about nanodroplets at the beginning of the paper when the main point of the work is about broad wavelength operation multi-function optical logic.*

Alternatively, another approach would be to give the section explaining measurements on characterizing the nanodroplets (the section with Fig 1) a different name, like “Characterization of Nanodroplets.” This way the reader would know that this is specialized information about part of the system.

Our Reply:

Many thanks for these kind suggestions. We have tried to move the whole part of nanodroplet characterization to the supplementary information or re-organize the description order of the main text by placing the nanodroplet characterization behind the device structure. However, we found that would probably arise more confusion

and/or misunderstanding on the audience because the underlying mechanism of the device is based on the nanodroplet's non-trivial broadband plasmonic nonlinearity for generation of thermal-induced nonlinear refractive index field (RIF). If the nanodroplet properties are not introduced first, it is difficult to clearly illustrate why the logical functions can be enabled by the dynamic RIF modulation and how to implement reversible spatial-phase shift to switch the nine logic gates in one configuration.

Therefore, we have done our best to shorten the characterization part and name it as “**Characterization of Nanodroplets**” according to the Reviewer’s great advice. The following parts of logical device are also divided into “**Structure and mechanism of all-optical logic gates**” and “**Implementation of reconfigurable all-optical logic functions**”.

Corresponding revisions:

Line 19 in Page 4: Characterization of Galinstan Nanodroplets

Line 22 in Page 7: Structure and mechanism of all-optical logic gate

Line 1 in Page 12: Implementation of reconfigurable all-optical logic functions

Comment 5: References: *The statement, “The past decade has witnessed vast advancements AOLGs through linear or nonlinear optical modulation of nano-semiconductors, ranging from stimulated scattering and photoluminescence (PL) of nanowires or nanospheres¹⁰⁻¹² to spatial self-phase modulation (SSPM) or spatial cross-phase modulation (SXPM) of nanosheets¹³⁻¹⁵” is very under-referenced. More should be included. I would recommend the authors look at two of the references they already cite in the main paper and the supplement:*

1) Minzioni, P. et al. Roadmap on all-optical processing. J. Opt. 21, 063001 (2019)

2) Maram, R. et al. Frequency-domain ultrafast passive logic: NOT and XNOR gates. Nat. Commun. 11, 5839 (2020).

The authors may be able to use some of the references in these works or forward cite appropriate references from these papers for current work across AOLGs. Both have a lot of review information on the field.

Note, in general, that optical logic gates fall into two main categories of operation principle: 1) nonlinearity and 2) interference. The list the authors cite in this section, refs 10-15 are really all nonlinear devices. This needs to be changed (i.e. add at least one reference that actually uses linear interference)

Note also that Maram et al’s work “Frequency-domain ultrafast passive logic” is expressly not a nonlinear technique- hence the word “passive” in the title. The authors incorrectly label this work as using “nonlinear fiber” in the chart in the supplement and so this should be changed. This work is an example of linear interference.

Our Reply: Thanks for the reviewer’s recommendation and correction.

(1) This statement has been revised as follows.

Line 22 in Page 3: The past decade has witnessed vast advancements of AOLGs based on linear or nonlinear optical modulation, ranging from stimulated scattering and photoluminescence (PL) of nanowires or nanospheres⁷⁻⁹, **linear interference with phase-only linear filtering**^{1,10}, to spatial self-phase modulation (SSPM) or spatial cross-phase modulation (SXPM) of nanosheets¹²⁻¹⁴.

[1] Maram, R. et al. Frequency-domain ultrafast passive logic: NOT and XNOR gates. *Nat. Commun.* **11**, 5839 (2020).

[10] Kaushal, S. et al. All-fibre phase filters with 1-GHz resolution for high-speed passive optical logic processing. *Nat. Commun.* **14**, 1808 (2023).

(2) We deeply apologize for mislabeling Maram et al’s work in **Table S4 (Table R8)**. It has been corrected to “**Phase-only linear filtering for linear interference**”.

Table R8 (Table S4) Comparison of Galinstan AOLG with representative optical logic-gate schemes

Platform	Functionality	Wavelength (nm)	Ref.
Gold – graphene – gold heterostructure	NOR/AND/OR/NAND	827	14
Semiconductor nanowire	NAND	458	15
Semiconductor nanowire	AND/OR/NAND	658	16
Polariton in Semiconductor microcavity	AND/OR	—	17
Ag nanowire	NOR/OR/NOT	633	18
Phase-only linear filtering for linear interference	NOT/XNOR	1542-1552	19
Perovskite photodetector	AND/OR/NAND/NOR/ NOT	400-1000	20
Rb atoms	NOT	479, 780	21
Graphene	NOR/OR/AND	1560–2100 nm	22
Semiconductor nanowire	AND/OR/NAND/NOR	532-890	23
Bulk silica and monolayer MoS ₂	AND/OR/XOR/XNOR/ NOR/NAND	451-1036	24
Liquid crystal	AND/OR/NOT	532	25

Plasmonic gratings for linear interference	AND/OR/NOT/NAND/ NOR/XOR/XNOR	600-700	26
Metal structure	AND/OR/NOT/NAND/ NOR/XNOR	700-930	27
Organic nanosphere	OR	405, 350-390, 400-440	28
Black arsenic-phosphorus	OR	532, 671	29
Semiconductor optical amplifier	AND/OR/NOT/NOR/X NOR	1549.3-1557.3	30
Polariton in Semiconductor microcavity	XNOR	—	31
Boron nanosheets	OR	457-671	32
MoSe ₂ nanoflakes	OR	405-671	33
Semiconductor optical amplifier	XOR/NOR/OR/NAND	1531.05-1553.79	34
Nonlinear fiber	AND/NOR/XOR/XNOR	1527.2-1565.6	35
All-fibre phase filters for linear interference	NOT/XNOR	1550	36
Liquid-metal nanodroplet	AND/OR/NOT/NOR/N AND/XNOR/XOR/IMP/ NIMP	532-2000 (400-4000 in potential)	This work

Comment 7: *The English needs to be tidied up a bit. It's mostly fine and consists mainly of a handful of small errors such as subject verb agreement (easily corrected in copyediting). However, there are some statements that were confusing to me or are particularly important to edit.*

(1) *Title: "Liquid-metal plasmon nonlinearity lights large-bandwidth all-in-one all-optical logic gates" I don't understand the use of the word, "lights" here. Perhaps change the title to, "All-in-one, all-optical logic gates using liquid metal plasmon nonlinearity";*

(2) *Abstract: change "spital" to "spatial";*

(3) *Introduction: I don't understand the following statement. In addition, due to the diffraction limitation of each optical component, it is almost impractical for large scale optical integration by simply packaging different logic gates on a chip as widely employed in electric circuits. Otherwise, complex integration of optical processors by diverse AOLG elements is bulky and high energy-consuming. Therefore, extra rational strategies for compact multifunction-integrated and broadband optical logic operators have been urgently sought."*

What is it about diffraction that limits each component? How does bulk lead to

energy consumption? Also, there are a handful of awkward modifiers, “extra rational” “almost impractical,” “urgently sought.” These phrases should be changed. This statement seems to be saying that different optical logic operators will need to be made physically different. This will give different spatial modes or entail many components having different spatial mode requirements, now making each component bulky, unique and not easily printable in a universal way like electronics. Is that what this is trying to say?

(4) *Body:* The word phrase “preciously investigate” in line 252 is a bit bizarre. Just remove the word “precious.”

Our Reply:

We are very grateful to reviewer for these crucial corrections. With the help of your valuable suggestion, we made a lot of modifications as follows.

(1) The title has been changed to “All-in-one, all-optical logic gates using liquid metal plasmon nonlinearity”

(2) Abstract: “reversible **spital**-phase shift” has been revised to “reversible **spatial**-phase shift”.

(3) We mean that the diffraction limit of light is a fundamental obstacle for reducing the dimensions of optical logic components to the length scales of electronic devices in integrated circuits [*Nature*, 565, 560 (2018); *Nat. Commun.* 2, 387 (2011)].

We do agree with the reviewer’s more concise expression, and the corresponding statement has been revised as follow.

Line 18 in Page 3: The practicality of optical processor implies the demand of multifunctional AOLG components since different optical logic gates will need to be made physically different. This will give different spatial modes or entail many components having different spatial mode requirements, now making each component bulky, unique, and not easily printable in a universal way like electronics.

Other revisions:

(i) The sentence “it is **almost impractical** for large scale optical integration...” has been deleted.

(ii) **Line 28 in page 3:** the sentence “Therefore, **extra rational** strategies for compact multifunction-integrated and broadband optical logic operators have been **urgently sought**” has been revised to “**In this case, more rational strategies have been long pursued** for the further development towards high scalability and multifunctionality.”

(4) **Line 10 in page 13:** the sentence “As ultrafast transient absorption (TA)

spectroscopy has been widely used to **precisely investigate...**” has been revised to “As ultrafast transient absorption (TA) spectroscopy has been widely used to **investigate...**”

Additionally, the English has been carefully polished throughout the main text and the supplementary file.

Based on all the constructive comments as well as valuable suggestions, We hope these detailed explanations and modifications above can essentially improve the quality of our manuscript.

Reviewer #1 (Remarks to the Author):

After reading the responses from the authors and checking the revised manuscript, I admire that the authors tried their best addressing my questions well. The newly added results and analysis significantly reinforce the paper, I think they are solid and sounding. Moreover, I also see the author have added more discussions about the significance and practical potential of this work, it's nice for broadening the audience. In sum, I can recommend to publish this research in Nature Communications.

Reviewer #2 (Remarks to the Author):

The revised version can be accepted for publication.